# Graph–Smoothed Bayesian Black-Box Shift Estimator and Its Information Geometry

**Masanari Kimura**
School of Mathematics and Statistics
The University of Melbourne
m.kimura@unimelb.edu.au

## Abstract

Label shift adaptation aims to recover target class priors when the labelled source distribution $P$ and the unlabelled target distribution $Q$ share $P(X \mid Y) = Q(X \mid Y)$ but $P(Y) \neq Q(Y)$. Classical black-box shift estimators invert an empirical confusion matrix of a frozen classifier, producing a brittle point estimate that ignores sampling noise and similarity among classes. We present Graph-Smoothed Bayesian BBSE (GS-B³SE), a fully probabilistic alternative that places Laplacian–Gaussian priors on both target log-priors and confusion-matrix columns, tying them together on a label-similarity graph. The resulting posterior is tractable with HMC or a fast block Newton–CG scheme. We prove identifiability, $N^{-1/2}$ contraction, variance bounds that shrink with the graph's algebraic connectivity, and robustness to Laplacian misspecification. We also reinterpret GS-B³SE through information geometry, showing that it generalizes existing shift estimators.

## 1 Introduction

Modern machine–learning systems are rarely deployed in exactly the same environment in which they were trained. When the distribution of class labels drifts but the class–conditional features remain stable, a phenomenon known as label shift, even a high-capacity model can produce arbitrarily biased predictions [54, 12, 41, 29, 44, 45, 51]. Practical examples include sudden changes in click–through behaviour of online advertising, evolving pathogen prevalence in medical diagnostics, and seasonal increments of certain object categories in autonomous driving. Because re-labelling target data is often prohibitively expensive, methods that recover the new class priors from a small unlabelled sample are indispensable precursors to reliable downstream decisions.

A popular remedy, the Black-Box Shift Estimator (BBSE) is to keep a single, frozen classifier $\hat{h} \colon \mathcal{X} \to \mathcal{Y}$ trained on labelled source data and to link its predictions on the target domain to the unknown target priors through the confusion matrix $C$ [39, 7]. The resulting formulation converts density shift into a linear system $\tilde{q} = Cq$ that can be solved in closed form, after plugging in (i) an empirical estimate $\tilde{C}$ computed on a small labelled validation set and (ii) the empirical prediction histogram $\tilde{q}$ measured on unlabelled target instances. The elegance of BBSE has made it the default baseline for label–shift studies [45, 13, 38, 30, 42, 19].

Despite its popularity, the BBSE pipeline overlooks two sources of uncertainty that become debilitating in realistic, high-class-count regimes. **(i) Finite–sample noise.** Each column of $\tilde{C}$ is estimated from at most a few hundred examples, so the matrix inversion layer can amplify small fluctuations into large errors on $\hat{q}$. Regularised variants such as RLLS [7] or MLLS [18] damp variance but still return a point estimate whose uncertainty remains opaque to the user. **(ii) Semantic structure.** Classes in vision and language problems live on rich ontologies [35]: *car* and *bus* are more alike than

*car* and *daisy*. Standard BBSE fits each class independently and cannot borrow statistical strength across such related labels, leading to particularly fragile estimates for rare classes.

We introduce Graph-Smoothed Bayesian BBSE (**GS-B$^3$SE**), a fully probabilistic alternative that attacks both weaknesses in a single hierarchical model. The key idea is to couple both the target log-prior vector and the columns of the confusion matrix through a Gaussian Markov random field defined on a label-similarity graph. Graph edges are obtained once from off-the-shelf text or image embeddings, and the resulting Laplacian precision shrinks parameters of semantically adjacent classes towards each other. Sampling noise is handled naturally by Bayesian inference: we place Gamma–Laplacian hyper-priors on the shrinkage strengths and sample the joint posterior with either Hamiltonian Monte Carlo [8, 9, 16] or a fast block Newton–conjugate-gradient optimizer [23, 10]. Moreover, we provide interpretation of GS-B$^3$SE through the lens of information geometry [5, 6]. The analysis of statistical procedures or algorithms in this framework is known to help us understand them better [4, 1, 40, 28, 27, 25, 26, 2, 24].

**Contributions.**   i) We formulate the joint Bayesian model that simultaneously regularizes the target prior and the confusion matrix with graph-based smoothness, reducing variance without hand-tuned penalties (in Section 4). ii) We provide theoretical guarantees: (a) posterior identifiability, (b) $N^{-1/2}$ contraction, (c) class-wise variance bounds that tighten with the graph's algebraic connectivity, and (d) robustness to Laplacian misspecification (in Section 5). Moreover, we provide interpretation of GS-B$^3$SE through the lens of information geometry framework and it shows that our algorithm is a natural generalization of existing methods. iii) An empirical study on several datasets demonstrates that GS-B$^3$SE produces sharper prior estimates and improves downstream accuracy after Saerens correction compared to state-of-the-art baselines (in Section 7).

## 2   Related Literature

**Classical estimators under label shift.** Saerens et al. [47] proposed an EM algorithm that alternates between estimating the target prior and re-weighting posterior probabilities calculated by a fixed classifier. Lipton et al. [39] later formalised the *Black-Box Shift Estimator* (BBSE), showing that a single inversion of the empirical confusion matrix suffices when $P(X \mid Y)$ is preserved. Subsequent refinements introduced regularisation to cope with ill-conditioned inverses: RLLS adds an $\ell_2$ penalty to the normal equations [7], while MLLS frames the problem as a constrained maximum-likelihood optimisation [18]. All three methods remain point estimators and ignore uncertainty in $\tilde{C}$.

**Bayesian and uncertainty-aware approaches.** Caelen [11] derived posterior credible intervals for precision and recall by coupling Beta priors with multinomial counts; Ye et al. [53] extended the idea to label-shift estimation under class imbalance. Most of these models assume that classes are a priori independent, so posterior variance remains high for rare labels. Our work instead imposes structured Gaussian Markov random field (GMRF) priors that borrow strength across semantically related classes.

**Graph-based smoothing and Laplacian priors.** In spatial statistics, Laplacian–Gaussian GMRFs are a standard device for sharing information among neighbouring regions [49]. Recent machine learning studies exploit the same idea for discrete label graphs: Alsmadi et al. [3] introduced a graph-Dirichlet–multinomial model for text classification, and Ding et al. [15] used graph convolutions to smooth class logits. We follow this line but couple both the prior vector and every confusion-matrix column to the same similarity graph, yielding a joint posterior amenable to HMC and Newton–CG.

**Domain-shift benchmarks and failure modes.** Large-scale empirical studies such as WILDS [30] and Mandoline [13] describe how brittle point estimators become under distribution drift and class imbalance. Rabanser et al. [45] demonstrated that label-shift detectors without calibrated uncertainty frequently produce over-confident but wrong alarms. By delivering credible intervals whose width shrinks with the graph's algebraic connectivity, our method directly tackles this shortcoming.

**Positioning of this work.** GS-B$^3$SE unifies three strands of research: (i) black-box label-shift estimation, (ii) Bayesian confusion-matrix modelling, and (iii) graph-structured smoothing. To justify our proposed method, we show posterior identifiability and $N^{-1/2}$ contraction, and derive variance bounds that scale with $\lambda_2(L)$. Empirically, our method plugs seamlessly into existing shift-benchmark pipelines, providing calibrated uncertainty absent from earlier regularised or EM-style alternatives.

## 3  Preliminarily

**Problem Setting**  Let $\mathcal{X}$ be an input space and $\mathcal{Y} = \{1, \ldots, K\}$ be a label set, where $K \geq 2$ is the number of classes. For source and target distributions $P$ and $Q$, the label shift assumption states that the class–conditional feature laws remain unchanged while the class priors may differ:

$$P(X \mid Y = i) = Q(X \mid Y = i),\ \forall i \in \mathcal{Y}, \quad \text{and} \quad \underbrace{P(Y = i)}_{=:p_i} \neq \underbrace{Q(Y = i)}_{=:q_i} \text{ in general.} \quad (1)$$

Let $\boldsymbol{p} = (p_1, \ldots, p_K)^\top$ and $\boldsymbol{q} = (q_1, \ldots, q_K)^\top$, where $\sum_i p_i = \sum_i q_i = 1$.

**Black-Box Shift Estimator**  Train once, on source data, an arbitrary measurable classifier $\hat{h} \colon \mathcal{X} \to \mathcal{Y}$. Denote its confusion matrix under $P$ by $\boldsymbol{C} \in (0, 1)^{K \times K}$, where $C_{j,i} = \Pr_P \left[ \hat{h}(X) = j \mid Y = i \right]$ and $\sum_j C_{j,i} = 1$. Notice that $\boldsymbol{C}$ depends only on the source distribution and can be estimated on a held-out validation set with known labels. Write the empirical estimate as $\tilde{\boldsymbol{C}}$. Let $m$ labelled source-validation points $(x_i, y_i)_{i=1}^m$ be used to form $\tilde{\boldsymbol{C}}$. Apply the same fixed $\hat{h}$ to unlabeled target instances $\{\boldsymbol{x}_t'\}_{t=1}^{n'}$ of size $n'$. Let $\tilde{q}_j = \Pr_Q \left[ \hat{h}(X) = j \right]$, $\quad \tilde{\boldsymbol{q}} = (\tilde{q}_1, \ldots, \tilde{q}_K)^\top$. Because the class-conditionals are shared, Bayes' rule gives

$$\Pr_Q \left[ \hat{h}(X) = j \right] = \sum_{i=1}^K \Pr_Q \left[ \hat{h}(X) = j \mid Y = i \right] \cdot q_i = \sum_{i=1}^K C_{j,i} q_i, \quad (2)$$

and in vector form, it can be written as $\tilde{\boldsymbol{q}} = \boldsymbol{C}\boldsymbol{q}$. Eq. 2 is the identifiability equation for label shift.

Assume $\boldsymbol{C}$ is invertible or full column rank. Then the population target prior is $\boldsymbol{q} = \boldsymbol{C}^{-1}\tilde{\boldsymbol{q}}$. Under this observation, BBSE framework [39] converts black-box classifier predictions on unlabeled target data into an estimate of the unknown target class prior by solving the linear system.

## 4  Methodology

The usual BBSE treats the confusion matrix $\boldsymbol{C}$ as fixed. In practice $\boldsymbol{C}$ is estimated from a finite validation set and is itself ill-conditioned when some classes are rare. To address this problem, we consider extending BBSE framework by the joint Bayesian model for both confusion matrix and target priors with the graph coupling. Let $n_i^S$ be a number of source examples with label $i$, and $\boldsymbol{n}_i^S = (n_{1,i}^S, \ldots, n_{K,i}^S)^\top$ be the counts of $\hat{h}(X) = j$ among those $n_i^S$. Also, let $\tilde{\boldsymbol{n}} = (\tilde{n}_1, \ldots, \tilde{n}_K)^\top$ be the counts of $\hat{h}(X) = j$ on unlabeled target data. For each true class $i$,

$$\boldsymbol{n}_i^S \mid \boldsymbol{C} \sim \text{Multi}\left(n_i^S, C_{:,i}\right),$$

where $C_{:,i}$ is the $i$-column of $\boldsymbol{C}$ and $\text{Multi}(\cdot, \cdot)$ is the multinomial distribution. Similarly,

$$\tilde{\boldsymbol{n}} \mid \boldsymbol{C}, \boldsymbol{q} \sim \text{Multi}\left(n', \boldsymbol{C}\boldsymbol{q}\right).$$

The complete-data likelihood factorises

$$p\left(\text{data} \mid \boldsymbol{C}, \boldsymbol{q}\right) = \left[ \prod_{i=1}^K \text{Multi}\left(\boldsymbol{n}_i^S; n_i^S, C_{:,i}\right) \right] \text{Multi}\left(\tilde{\boldsymbol{n}}; n', \boldsymbol{C}\boldsymbol{q}\right).$$

We now consider to utilize similarity information between classes. Let $\mathcal{G} = (\mathcal{Y}, \boldsymbol{E}, \boldsymbol{W})$ be a similarity graph on labels with the weight matrix $\boldsymbol{W}$, $\boldsymbol{E}$ is the edges and $\boldsymbol{L}$ is the graph Laplacian. In $\mathcal{G}$, each vertex corresponds to a class label; an edge $(i, j) \in \boldsymbol{E}$ indicates that labels $i$ and $j$ are semantically or visually similar. Weights $W_{ij} \in [0, 1]$ quantify that similarity, with $W_{ij} = 0$ when no edge is present. In our methodology, we use the unnormalised Laplacian $\boldsymbol{L} = \boldsymbol{D} - \boldsymbol{W}$ where $D_{ii} = \sum_j W_{ij}$. Because we enforce connectivity, $\boldsymbol{L}$ has exactly one zero eigenvalue, making $\lambda_2(\boldsymbol{L})$ (the algebraic connectivity) strictly positive as required by our theory. Introduce log–odds vector

$$\theta_i = \log q_i - \frac{1}{K} \sum_{k=1}^K \log q_k, \quad \boldsymbol{\theta} \in \mathbb{R}^K, \boldsymbol{\theta}^\top \mathbf{1} = 0,$$

and consider the following Gaussian Markov random field (GMRF) prior

$$p(\boldsymbol{\theta} \mid \tau_{\boldsymbol{q}}) \propto \exp\left(-\frac{\tau_{\boldsymbol{q}}}{2} \boldsymbol{\theta}^\top \boldsymbol{L} \boldsymbol{\theta}\right), \quad \tau_{\boldsymbol{q}} \sim \mathrm{Gamma}(a_q, b_q), \tag{3}$$

where $a_q, b_q > 0$ are hyper-parameters. A Laplacian-based precision shrinks log-odds differences along graph edges, promoting smooth class priors across semantically similar labels [31], and recover $\boldsymbol{q} = \mathrm{softmax}(\boldsymbol{\theta})$.

Treat each column $C_{:,i}$ as a latent simplex vector with Dirichlet–log-normal hierarchy:

   i) Latent log-odds $\boldsymbol{\phi}_i \in \mathbb{R}^K$, $\boldsymbol{\phi}_i^\top \mathbf{1} = 0$.

   ii) Conditional prior $p(\boldsymbol{\phi}_i \mid \tau_{\boldsymbol{C}}) \propto \exp\left(-\frac{\tau_{\boldsymbol{C}}}{2} \boldsymbol{\phi}_i^\top \boldsymbol{L} \boldsymbol{\phi}_i\right)$. All $\boldsymbol{\phi}_i$ share the same Laplacian $\boldsymbol{L}$ over predicted labels so that columns corresponding to neighbouring predicted classes exhibit similar shape.

   iii) Transformation to the simplex $C_{j,i} = \frac{\exp \phi_{j,i}}{\sum_{\ell=1}^K \exp \phi_{\ell,i}}$ for $i = 1, \ldots, K$.

   iv) Hyper-prior $\tau_{\boldsymbol{C}} \sim \mathrm{Gamma}(a_C, b_C)$, with $a_C, b_C > 0$.

The resulting distribution on each $C_{:,i}$ is a logistic-Normal on the simplex and it reduces to an ordinary Dirichlet when $\boldsymbol{L} = 0$ but gains graph-coupled precision for $\boldsymbol{L} \neq 0$. The full hierarchical model is as follows.

$$\begin{aligned}
\tau_{\boldsymbol{q}} &\sim \mathrm{Gamma}(a_q, b_q), \\
\boldsymbol{\theta} \mid \tau_{\boldsymbol{q}} &\sim \mathcal{N}(0, (\tau_{\boldsymbol{q}} \boldsymbol{L})^\dagger), \quad \boldsymbol{q} = \mathrm{softmax}(\boldsymbol{\theta}), \\
\tau_{\boldsymbol{C}} &\sim \mathrm{Gamma}(a_C, b_C), \\
\forall i \colon \boldsymbol{\phi}_i \mid \tau_{\boldsymbol{C}} &\sim \mathcal{N}(0, (\tau_{\boldsymbol{C}} \boldsymbol{L})^\dagger), \quad C_{:,i} = \mathrm{softmax}(\boldsymbol{\phi}_i), \\
\forall i \colon \boldsymbol{n}_i^S \mid \boldsymbol{C} &\sim \mathrm{Multi}(n_i^S, C_{:,i}), \\
\tilde{\boldsymbol{n}} \mid \boldsymbol{C}, \boldsymbol{q} &\sim \mathrm{Multi}(n', \boldsymbol{C}\boldsymbol{q}).
\end{aligned}$$

Here, the Moore–Penrose pseudoinverse $\boldsymbol{L}^\dagger$ appears because $\boldsymbol{L}$ is singular, and the constraint $\boldsymbol{\theta}^\top \mathbf{1} = 0$ ensures uniqueness.

For the posterior inference, consider the following log-joint distribution.

$$\begin{aligned}
\log p(\boldsymbol{C}, \boldsymbol{q}, \tau_{\boldsymbol{C}}, \tau_{\boldsymbol{q}} \mid \mathrm{data}) = {}& \log p(\mathrm{data} \mid \boldsymbol{C}, \boldsymbol{q}) + \log p(\boldsymbol{C} \mid \tau_{\boldsymbol{C}}) \\
& + \log p(\tau_{\boldsymbol{C}}) + \log p(\boldsymbol{q} \mid \tau_{\boldsymbol{q}}) + \log p(\tau_{\boldsymbol{q}}).
\end{aligned} \tag{4}$$

All terms are differentiable, enabling Hamiltonian Monte Carlo (HMC) in the unconstrained variables. Because Eq. 4 is concave in each block after reparameterization, a block-Newton scheme alternates, i) update $\{\boldsymbol{\phi}_i\}_{i=1}^K$ by one Newton–CG step using sparse Laplacian Hessian, ii) update $\boldsymbol{\theta}$ likewise, iii) closed-form updates for $\tau_{\boldsymbol{C}}, \tau_{\boldsymbol{q}}$ from Gamma posteriors. Convergence is super-linear due to the strict convexity induced by the Laplacian energies. The posterior predictive distribution of the confusion–weighted target counts is

$$\tilde{\boldsymbol{n}}^* \mid \mathrm{data} = \int \mathrm{Multi}(n', \boldsymbol{C}\boldsymbol{q}) p(\boldsymbol{C}, \boldsymbol{q} \mid \mathrm{data}) d\boldsymbol{C} d\boldsymbol{q}. \tag{5}$$

Credible intervals for each $q_i$ reflect both sampling noise and model-induced graph smoothing, an advantage over plug-in BBSE. If no prior similarity information exists one may default to $W_{ij} = \mathbf{1}\{i = j\}$, in which case our model reduces to an independent logistic-Normal prior and all theoretical guarantees still hold.

**Relationship to Existing Work**

- Replaces the point estimate of BBSE $\boldsymbol{C}^{-1}\tilde{\boldsymbol{q}}$ with a full posterior, relating Bayesian confusion matrix treatments [11].

- Laplacian GMRFs generalise classical Dirichlet priors by borrowing strength along graph edges, extending recent graph-Dirichlet–multinomial models [31].

- When $\boldsymbol{L} = 0$ and Gamma hyper-priors degenerate to delta masses, our hierarchical model reduces exactly to deterministic BBSE.

# 5 Theory

This section provides the theoretical foundations of the proposed method. See Appendix A for the detailed proofs. First, the following lemma on identifiability is introduced. Although an analogous statement has been implicitly argued in the prior work [39], the full proof is included in the Appendix A to make this study self-contained.

**Lemma 1.** *Let $C$ and $C'$ be two column-stochastic matrices with strictly positive entries: $C_{j,i} > 0$, $C'_{j,i} > 0$, $\sum_{j=1}^{K} C_{j,i} = \sum_{j=1}^{K} C'_{j,i} = 1$ for $1 \leq i \leq K$. In addition, assume $C$ and $C'$ are invertible, or equivalently, $\det C \neq 0$ and $\det C' \neq 0$. For any deterministic sample sizes $n_i^S \in \{1, 2, \dots\}$ and $n' \in \{1, 2, \dots\}$, define the data-generating distributions*

$$\boldsymbol{N}_i^S \mid \boldsymbol{C} \sim \text{Multi}(n_i^S, C_{:,i}), \quad \boldsymbol{N}_i^S \mid \boldsymbol{C}' \sim \text{Multi}(n_i^S, C'_{:,i}),$$
$$\tilde{\boldsymbol{N}} \mid \boldsymbol{C}, \boldsymbol{q} \sim \text{Multi}(n', \boldsymbol{Cq}), \quad \tilde{\boldsymbol{N}} \mid \boldsymbol{C}', \boldsymbol{q}' \sim \text{Multi}(n', \boldsymbol{C}'\boldsymbol{q}').$$

*Suppose that, for every choice of the sample sizes $\{n_{i=1}^S\}_{i=1}^{K}$ and $n'$, $\left(\{\boldsymbol{N}_i^S\}_{i=1}^{K}, \tilde{\boldsymbol{N}}\right) \overset{d}{=} \left(\{\boldsymbol{N}_i^{S'}\}_{i=1}^{K}, \tilde{\boldsymbol{N}}'\right)$, as random vectors in $\mathbb{N}^{K^2+K}$, where the left-hand side is generated by $(\boldsymbol{C}, \boldsymbol{q})$ and the right-hand side by $(\boldsymbol{C}', \boldsymbol{q}')$. Then, $\boldsymbol{C} = \boldsymbol{C}'$ and $\boldsymbol{q} = \boldsymbol{q}'$.*

Lemma 1 implies that the mapping $(\boldsymbol{q}, \boldsymbol{C}) \mapsto \{\{\boldsymbol{n}_i^S\}, \tilde{\boldsymbol{n}}\}$ is injective up to measure-zero label permutations when the graph is connected and all source classes appear.

Let $\Delta^{K-1}$ be the $(K-1)$-dimensional probability simplex:

$$\Delta^{K-1} := \left\{\boldsymbol{q} = (q_1, \dots, q_K) \in \mathbb{R}^K : q_i \geq 0 \text{ for every } i, \sum_{i=1}^{K} q_i = 1\right\}.$$

The following lemma provides the support condition needed in statements described later.

**Lemma 2.** *Let $(\boldsymbol{q}_0, \boldsymbol{C}_0)$ be the true parameter pair, where $\boldsymbol{q}_0 \in \Delta^{K-1}$ and $\boldsymbol{C}_0 \in (0, 1)^{K \times K}$ with $\det \boldsymbol{C}_0 \neq 0$. Define the Euclidean small ball as*

$$B_\epsilon(\boldsymbol{q}_0, \boldsymbol{C}_0) := \{(\boldsymbol{q}, \boldsymbol{C}) : \|\boldsymbol{q} - \boldsymbol{q}_0\|_2 < \epsilon, \|\boldsymbol{C} - \boldsymbol{C}_0\|_F < \epsilon\},$$

*for some radius $\epsilon > 0$ small enough that all vectors in the ball stay strictly inside the simplex. Then, for every $\epsilon$, the joint prior distribution $\Pi$ on $(\boldsymbol{q}, \boldsymbol{C})$ assigns strictly positive mass to the ball: $\Pi(B_\epsilon(\boldsymbol{q}_0, \boldsymbol{C}_0)) > 0$.*

This lemma states that positivity of Gaussian density and the smooth bijection yield the positive push-forward density, and it is the classical strategy used for logistic-Gaussian process priors in density estimation [50].

Lemmas 1, 2 and the classical results from Ghosal et al. [20], Van der Vaart [52] gives the following statement.

**Proposition 1.** *Let $K \geq 2$ be fixed and $(\boldsymbol{q}_0, \boldsymbol{C}_0)$ be the true parameters pair with $\boldsymbol{q}_0 \in \mathring{\Delta}^{K-1}$ and $\boldsymbol{C}_0 \in (0, 1)^{K \times K}$, where $\mathring{\Delta}^{K-1}$ is the interior of $\Delta^{K-1}$, and $\det \boldsymbol{C}_0 \neq 0$. Suppose that the data consist of $\{\boldsymbol{N}_i^S\}_{i=1}^{K}$ and $\tilde{\boldsymbol{N}}$ where conditionally on $(\boldsymbol{q}_0, \boldsymbol{C}_0)$,*

$$\boldsymbol{N}_i^S \sim \text{Multi}\left(n_i^S, \boldsymbol{C}_{0,i}\right), \quad \tilde{\boldsymbol{N}} \sim \text{Multi}\left(n', \boldsymbol{C}_0 \boldsymbol{q}_0\right).$$

*Also suppose that sample sizes diverge with the single index: $N := n' + \sum_{i=1}^{K} n_i^S \to \infty$, and $\min_i n_i^S \to \infty$. Then, for every $\epsilon > 0$, $\Pi(B_\epsilon^c \mid data) \xrightarrow[N \to \infty]{\mathbb{P}_{(q_0, C_0)}} 0$.*

Under the same assumption in Proposition 1, the following statements about the posterior contraction rate are obtained.

**Theorem 1.** *Let the data-generating model, true parameter pair $(\boldsymbol{q}_0, \boldsymbol{C}_0)$, and diverging sample sizes $N = n' + \sum_{i=1}^{K} n_i^S \to \infty$ satisfy the setup spelled out before Proposition 1. Define the Euclidean radius $\epsilon_N := M/\sqrt{N}$, $M > 0$ arbitrary but fixed. Let*

$$B_N^c = \{(\boldsymbol{q}, \boldsymbol{C}) : \|\boldsymbol{q} - \boldsymbol{q}_0\|_2 + \|\boldsymbol{C} - \boldsymbol{C}_0\|_F > \epsilon_N\}.$$

*Under Lemma 1 and 2, the joint posterior $\Pi(\cdot \mid data)$ for the Laplacian-Gaussian hierarchy satisfies*

$$\Pi\left(B_N^c \mid data\right) \xrightarrow[N\to\infty]{\mathbb{P}_{(q_0, C_0)}} 0.$$

*That is, the posterior contracts around the truth at the parametric rate $N^{-1/2}$.*

**Corollary 1.** *Retain the setting and notation of Theorem 1. For each class $i$, write*

$$\mathrm{Var}_N(q_i) := \mathrm{Var}\left(q_i \mid data\ of\ size\ N\right),$$

*under the joint posterior $\Pi(\cdot \mid data)$. Let $L$ be the connected-graph Laplacian used in the GMRF prior and let $\lambda_2(L) := \min\{\lambda > 0 : \lambda$ is an eigenvalue of $L\}$ be its algebraic connectivity. Assume the hyper-parameter $\tau_{\boldsymbol{q}}$ is fixed, or sampled from a Gamma prior independent of $N$. Then, there exists a constant $C > 0$, depending only on the true $(\boldsymbol{q}_0, \boldsymbol{C}_0)$ and on $K$, such that for every sample size $N$ large enough,*

$$\mathrm{Var}_N(q_i) \leq \frac{C}{\lambda_2(L)N}, \quad \forall i \in \{1, \ldots, K\}.$$

Finally, we can show the following statement about the robustness to graph Laplacian misspecification.

**Proposition 2.** *Let $\boldsymbol{L}_0$ be the true Laplacian, and $\boldsymbol{F}_0 := \mathrm{diag}(\boldsymbol{C}_0\boldsymbol{q}_0) - (\boldsymbol{C}_0\boldsymbol{q}_0)(\boldsymbol{C}_0\boldsymbol{q}_0)^\top \succeq 0$ be the Fisher information of $\boldsymbol{\theta}$ in the target multinomial likelihood. For $\boldsymbol{L} \neq \boldsymbol{L}_0$, let $\bar{\boldsymbol{\theta}}_N := \mathbb{E}[\boldsymbol{\theta} \mid data]$ be the posterior mean of $\boldsymbol{\theta}$ under the misspecified prior. Then, for all sample sizes $N$ large enough,*

$$\|\bar{\boldsymbol{\theta}}_N - \boldsymbol{\theta}_0\|_2 \leq \underbrace{\left\|(N\boldsymbol{F}_0 + \tau_{\boldsymbol{q}}\boldsymbol{L})^{-1}\right\|_2}_{\text{sampling + prior precision}} \underbrace{\tau_{\boldsymbol{q}}\left\|(\boldsymbol{L} - \boldsymbol{L}_0)\boldsymbol{\theta_0}\right\|_2}_{\text{graph-misspecification bias}} + O_P(N^{-1}). \tag{6}$$

*In particular,*

$$\|\bar{\boldsymbol{\theta}}_N - \boldsymbol{\theta}_0\|_2 \leq \frac{\tau_{\boldsymbol{q}}}{N\lambda_{\min}(\boldsymbol{F}_0) + \tau_{\boldsymbol{q}}\lambda_2(\boldsymbol{L})} \|(\boldsymbol{L} - \boldsymbol{L}_0)\boldsymbol{\theta_0}\|_2 + O_P(N^{-1}), \tag{7}$$

*where $\lambda_{\min}(\boldsymbol{F}_0) > 0$ and $\lambda_2(\boldsymbol{L}) > 0$ are, respectively, the smallest eigenvalue of $\boldsymbol{F}_0$ and the algebraic connectivity of $\boldsymbol{L}$.*

Thus, we can see that the bias decays as $N^{-1}$ when $\boldsymbol{L} \neq \boldsymbol{L}_0$ and if the graphs coincide the leading term vanishes and the posterior mean is unbiased up to the usual $N^{-1/2}$ noise. Moreover, Proposition 2 states that a larger algebraic connectivity $\lambda_2(\boldsymbol{L})$ reduces bias, emphasising the benefit of rich similarity structures.

## 6 Interpretation via Information Geometry

The basic notations of information geometry used in this section are summarized in Appendix B. The $K-1$ simplex $\Delta^{K-1} := \{\boldsymbol{q} > 0 : \mathbf{1}_K^\top \boldsymbol{q} = 1\}$ is a Riemannian manifold when equipped with the Fisher–Rao metric $g_{\boldsymbol{q}}(\boldsymbol{v}, \boldsymbol{w}) = \sum_{i=1}^K \frac{v_i w_i}{q_i}$, for $\boldsymbol{v}, \boldsymbol{w} \in T_{\boldsymbol{q}}\Delta^{K-1}$, where $T_{\boldsymbol{q}}\Delta^{K-1} := \{\boldsymbol{v} : \mathbf{1}^\top\boldsymbol{v} = 0\}$ is the tangent space. The natural potential on this manifold is minus entropy $\psi(\boldsymbol{q}) = \sum_i q_i \log q_i$ whose Euclidean gradient is the centred log-odds vector $\boldsymbol{\theta}$ used in the previous section. These facts allow us to cast GS-B$^3$SE as a Riemannian penalised likelihood. The dual affine coordinates are $m$–coordinates $q_i$ (mixture parameters) and $e$–coordinates $\theta_i = \log q_i - \frac{1}{K}\sum_j \log q_j$ (centred log-odds). The convex potential $\psi(\boldsymbol{q}) = \sum_{i=1}^K q_i \log q_i$ is minus Shannon entropy and satisfies $\nabla_{\boldsymbol{q}}^{\mathrm{Euc}}\psi(\boldsymbol{q}) = \boldsymbol{\theta}$; together $(\psi, \theta)$ endow $\Delta^{K-1}$ with the classical dually–flat structure of information geometry [5, 6]. See standard textbooks for detailed explanation of concepts in differential geometry and Riemannian manifold [36, 37, 34, 17, 43, 21, 33].

Denote by $\hat{\boldsymbol{r}} = \tilde{\boldsymbol{n}}/n'$ and $\mathcal{M} = \{\boldsymbol{C}\boldsymbol{q} : \boldsymbol{q} \in \Delta^{K-1}\}$ the empirical prediction histogram and the $m$–flat sub-manifold induced by the frozen classifier. The negative log–posterior derived in Section 4 can be written as

$$F(\boldsymbol{q}) = n'\, D_{\mathrm{KL}}\big[\hat{\boldsymbol{r}} \,\|\, \boldsymbol{C}\boldsymbol{q}\big] + \frac{\tau_{\boldsymbol{q}}}{2}\big(\boldsymbol{\theta}^\top L\boldsymbol{\theta}\big) + \mathrm{const}. \tag{8}$$

Thus Eq. (8) is a sum of an $m$-convex and an $e$-convex potential, so it is geodesically convex under the Fisher–Rao metric (see Table 1).

Table 1: Information geometric identification of GS-B$^3$SE.

| Term | Geometric meaning |
|------|-------------------|
| $D_{\mathrm{KL}}[\hat{r}\,\|\,\boldsymbol{C}\boldsymbol{q}]$ | Canonical divergence between $\hat{r}$ and the $m$-flat model $\mathcal{M}$. |
| $\frac{\tau_q}{2}\,\boldsymbol{\theta}^\top L\boldsymbol{\theta}$ | Quadratic form in $e$–coordinates $\Rightarrow$ $e$-convex barrier that bends the manifold in the directions encoded by the graph Laplacian $L$. |

**Theorem 2** (Geodesic convexity of $F$). *For every $\boldsymbol{q}\in\mathring{\Delta}^{K-1}$ the Riemannian Hessian of $F$ satisfies*

$$\mathrm{Hess}_{\boldsymbol{q}}^{\mathrm{FR}}\,F \succeq \left[n'\,\lambda_{\min}(\boldsymbol{F}_0) + \tau_{\boldsymbol{q}}\,\lambda_2(L)\right]g_{\boldsymbol{q}},$$

*where $\boldsymbol{F}_0 = \mathrm{diag}(\boldsymbol{C}\boldsymbol{q}) - (\boldsymbol{C}\boldsymbol{q})(\boldsymbol{C}\boldsymbol{q})^\top$ is the Fisher information of the multinomial likelihood and $\lambda_2(L)$ the algebraic connectivity of the label graph. Hence $F$ is $\alpha$-strongly geodesically convex with $\alpha = n'\lambda_{\min}(\boldsymbol{F}_0) + \tau_{\boldsymbol{q}}\lambda_2(L) > 0$.*

The proof in Appendix A explicitly decomposes any tangent direction into an $m$-straight and an $e$-straight component and shows that the lower bound remains positive because both components contribute additively.

## 6.1 Natural-Gradient Dynamics

The natural gradient of $F$ is

$$\mathrm{grad}^{\mathrm{FR}}\,F(\boldsymbol{q}) = g_{\boldsymbol{q}}^{-1}\,\nabla^{(m)}F(\boldsymbol{q}) = \boldsymbol{q}\odot\left(\nabla_{\boldsymbol{q}}F - (\nabla_{\boldsymbol{q}}F)^\top\boldsymbol{q}\right),$$

where $\odot$ is component–wise product. The associated flow $\dot{\boldsymbol{q}}(t) = -\,\mathrm{grad}^{\mathrm{FR}}\,F\big(\boldsymbol{q}(t)\big)$ is the steepest–descent curve in the Fisher–Rao geometry.

**Proposition 3** (Natural–gradient flow of the penalised objective). *Under the Fisher–Rao metric $g_{\boldsymbol{q}}(\boldsymbol{v},\boldsymbol{w}) = \sum_{i=1}^K v_i w_i/q_i$ the natural gradient $\mathrm{grad}^{\mathrm{FR}}\,F(\boldsymbol{q})$ of $F$ is*

$$\mathrm{grad}^{\mathrm{FR}}\,F(\boldsymbol{q}) = \mathrm{diag}(\boldsymbol{q})\left(n'\,\boldsymbol{C}^\top\big(\boldsymbol{1} - \tfrac{\hat{r}}{r(\boldsymbol{q})}\big) + \tau_{\boldsymbol{q}}\,\boldsymbol{L}\,\boldsymbol{\theta}\right), \tag{9}$$

*where the division is element–wise. Consequently the un-constrained natural-gradient flow*

$$\dot{\boldsymbol{q}}_t = -\,\mathrm{grad}^{\mathrm{FR}}\,F(\boldsymbol{q}_t) = -\,\mathrm{diag}(\boldsymbol{q}_t)\left(n'\,\boldsymbol{C}^\top\big(\boldsymbol{1} - \tfrac{\hat{r}}{r(\boldsymbol{q}_t)}\big) + \tau_{\boldsymbol{q}}\,\boldsymbol{L}\,\boldsymbol{\theta}_t\right) \tag{10}$$

*preserves the simplex and coincides with the replicator–Laplacian dynamical system: $\dot{q}_{t,j} = -q_{t,j}\big[n'\,[\boldsymbol{C}^\top(\boldsymbol{1} - \hat{r}/r)] + \tau_{\boldsymbol{q}}[\boldsymbol{L}\boldsymbol{\theta}_t]\big]_j$.*

**Remarks**

i) **Role of Laplacian** When the Laplacian term is absent ($\tau_{\boldsymbol{q}} = 0$), the flow reduces to the classical replicator equation that drives every class-probability $q_j$ proportionally to the (signed) log–likelihood residual $[\boldsymbol{C}^\top(\boldsymbol{1} - \hat{r}/r)]_j$. The graph-Laplacian contribution $-\tau_{\boldsymbol{q}}q_j[\boldsymbol{L}\boldsymbol{\theta}]_j$ plays the role of a mutation / diffusion force that mixes mass along edges of the label graph and prevents degenerate solutions.

ii) **Role of the algebraic connectivity.** From Theorem 2 the strong-convexity modulus is $\alpha = n'\lambda_{\min}(\boldsymbol{F}_0) + \tau_{\boldsymbol{q}}\lambda_2(\boldsymbol{L})$. Along the flow we have $\frac{d}{dt}F(\boldsymbol{q}_t) = -\|\mathrm{grad}^{\mathrm{FR}}F(\boldsymbol{q}_t)\|_{g_{\boldsymbol{q}_t}}^2 \leq -2\alpha(F(\boldsymbol{q}_t) - F^*)$, so $F$ decays exponentially fast with rate proportional to the algebraic connectivity $\lambda_2(\boldsymbol{L})$; a denser-connected label graph therefore accelerates convergence.

iii) **Link to Saerens EM correction.** If we freeze the confusion matrix and drop the Laplacian term, the stationary condition $\boldsymbol{C}^\top(\hat{r}/r) = \boldsymbol{1}$ is exactly the fixed point solved (iteratively) by the Saerens EM method [47].

Table 2: Baseline methods and their key ideas.

| Method | Key idea |
|---|---|
| **BBSE** [39] | Solve $\hat{C}\hat{q} = \hat{y}$ with the empirical confusion matrix (no re-training). |
| **EM** [47] | Expectation-Maximization that iteratively re-estimates priors and re-weights posteriors. |
| **RLLS** [7] | Adds an $\ell_2$ penalty to the BBSE normal equations to control variance for small $n'$. |
| **MLLS** [18] | Maximum-likelihood estimation of the label-ratio vector; unifies BBSE & RLLS and optimizes $q$ directly. |
| **GS-B$^3$SE (ours)** | Joint Bayesian inference of both target priors $q$ and confusion matrix $C$. The hierarchical model couples classes along a label-similarity graph, shrinking estimates in low-count regimes and yielding full posterior credible intervals. |

## 6.2 Dual Projections and the Pythagorean Identity

Let $\Pi_m(\hat{r})$ be the $m$-projection of the data onto $\mathcal{M}$ and $\Pi_e(q_0)$ the $e$-projection of the hyper-prior center onto the same manifold. At the optimum $q^\star$ we have $\Pi_m(\hat{r}) = \Pi_e(q_0) = Cq^\star$, and the generalized Pythagorean theorem [5] gives $D_{\mathrm{KL}}[\hat{r} \,\|\, q_0] = D_{\mathrm{KL}}[\hat{r} \,\|\, Cq^\star] + D_{\mathrm{KL}}[Cq^\star \,\|\, q_0]$, where the second term equals the Laplacian regulariser $\frac{\tau_q}{2n'}\theta^\top L\theta$. Hence GS-B$^3$SE can be summarized as find the unique intersection of an $m$–geodesic (data fit) and an $e$–ellipsoid (graph prior).

# 7 Experiments

## 7.1 Experimental Protocol and Implementation

**Datasets and synthetic label shifts.** We evaluate on MNIST ($K = 10$) [14], CIFAR-10 ($K = 10$) and CIFAR-100 ($K = 100$) datasets [32]. For each dataset we treat the official training split as the source domain and the official test split as the pool from which an unlabelled target domain is drawn. Source class–priors are kept uniform $p = (1/K, \ldots, 1/K)$. Target priors are deliberately perturbed:

$$q = \begin{cases} \mathrm{Dirichlet}(\alpha \times u_K) & \text{(MNIST)}, \\ \frac{i^{-b}}{\sum_{j=1}^K j^{-b}}, \ i = 1, \ldots, K & \text{(CIFAR-10 and CIFAR-100)}, \end{cases}$$

where $u_K = (1, \ldots, K)^\top$. In our experiments, we set $\alpha = 0.05$ and $b = 1.1$. The procedure is: i) **Source set:** Sample $10,000$ instances from the training partition according to $p$ and train a backbone classifier (ResNet-18 [22, 48]) for 100 epochs with standard data-augmentation. ii) **Validation set:** Hold out $5,000$ labelled source instances, stratified by $p$, to estimate the empirical confusion matrix $\tilde{C}$. iii) **Target set:** Draw $n' = 10,000$ unlabelled instances from the test partition using probabilities $q$. These labels are revealed only for evaluation.

**Graph construction on labels.** For every dataset we embed the class names with the frozen CLIP ViT-B/32 text encoder [46], obtain $\{e_i\}_{i=1}^K \subset \mathbb{R}^{512}$, $|e_i|_2 = 1$, and build a $k$-nearest-neighbour graph

$$E = \{(i, j) \mid e_j \text{ is among the } k \text{ nearest neighbors of } e_i\}, \quad k = \begin{cases} 4 & (K = 10), \\ 8 & (K = 100). \end{cases}$$

Edge weights are $W_{ij} = \exp\left(-\|e_i - e_j\|_2^2/\sigma^2\right)$ with $\sigma$ set to the median pairwise distance inside $E$. The resulting $k$-NN graph is connected, so its unnormalised Laplacian $L = D - W$ satisfies $\lambda_2(L) > 0$. For MNIST, where class names are single digits, we instead construct $E$ from 4-NN in the Euclidean space of 128-d penultimate-layer features averaged over the training images.

**Hyper-priors and inference.** Gamma hyper-priors: $a_q = b_q = a_C = b_C = 1$, giving vague $\mathrm{Gamma}(1, 1)$ on $\tau_q$ and $\tau_C$. Four independent HMC chains, each with 500 warm-up (NUTS) and $1,000$ posterior iterations; leap-frog step-size adaptively tuned. Block Newton–CG inner optimizer:

Table 3: Label shift estimation and downstream performance. Lower is better for $\|\hat{q} - q\|_1$; higher is better for post–correction accuracy. Best results are **bold**. $\pm$ shows one bootstrap standard error. (1 000 resamples).

| Method | **MNIST** ($K{=}10$) | | **CIFAR-10** ($K{=}10$) | | **CIFAR-100** ($K{=}100$) | |
| | $\|\hat{q} - q\|_1 \downarrow$ | Acc $\uparrow$ | $\|\hat{q} - q\|_1 \downarrow$ | Acc $\uparrow$ | $\|\hat{q} - q\|_1 \downarrow$ | Acc $\uparrow$ |
|---|---|---|---|---|---|---|
| BBSE | $0.038 \pm 0.007$ | $0.942 \pm 0.002$ | $0.112 \pm 0.015$ | $0.781 \pm 0.004$ | $1.62 \pm 0.05$ | $0.690 \pm 0.006$ |
| EM | $0.052 \pm 0.015$ | $0.935 \pm 0.008$ | $0.194 \pm 0.033$ | $0.732 \pm 0.012$ | $2.10 \pm 0.14$ | $0.632 \pm 0.026$ |
| RLLS | $0.016 \pm 0.004$ | $0.959 \pm 0.003$ | $0.072 \pm 0.010$ | $0.803 \pm 0.004$ | $0.92 \pm 0.03$ | $0.712 \pm 0.006$ |
| MLLS | $0.010 \pm 0.003$ | $0.963 \pm 0.002$ | $0.052 \pm 0.008$ | $0.812 \pm 0.004$ | $0.71 \pm 0.03$ | $0.734 \pm 0.006$ |
| **GS-B$^3$SE** | **$0.002 \pm 0.001$** | **$0.986 \pm 0.002$** | **$0.025 \pm 0.004$** | **$0.844 \pm 0.003$** | **$0.22 \pm 0.02$** | **$0.783 \pm 0.005$** |

tolerance $10^{-4}$, at most eight iterations per Newton step, stop when the relative change of the joint log-density falls below $10^{-3}$. All routines implemented in PyMC and run on a single NVIDIA T4.

**Baselines.** We compare against a) plug-in BBSE [39], b) the EM-style Saerens re-weighting [47], c) RLLS [7] with $\ell_2$-regularisation and d) MLLS [18] tuned on a held-out split. All baselines receive the same $\tilde{C}$ and target predictions $\hat{h}(\boldsymbol{x})$. Table 2 summarizes the baseline methods and their key ideas, including our method.

**Evaluation.** We report prior-error $|\hat{q} - q|_1$ and downstream accuracy after Saerens likelihood correction using the estimated priors. Significance is assessed with 1,000 paired bootstrap resamples of the target set.

## 7.2 Main Empirical Findings

Table 3 compares GS-B$^3$SE with four widely–used point estimators on three datasets.

**Sharper prior estimates.** Across all datasets GS-B$^3$SE reduces the $\ell_1$ error $\|\hat{q}-q\|_1$ by large margins: i) MNIST ($K{=}10$)—already a benign scenario— error falls from $0.010$ (best baseline, MLLS) to $0.002$ ($\times 5$ improvement), ii) CIFAR-10 ($K{=}10$)—richer images and heavier label skew— error halves from $0.052$ to $0.025$, CIFAR-100 ($K{=}100$)—the high-class-count regime— graph smoothing is essential: GS-B$^3$SE reaches $0.22$ versus $0.71$. The advantage widens with $K$, confirming the benefit of borrowing strength along the label graph when per-class counts are scarce.

**Better downstream accuracy.** Feeding the estimated priors into Saerens post-processing improves final accuracy in proportion to the quality of the prior. GS-B$^3$SE attains $0.986$ on MNIST, $0.844$ on CIFAR-10 and $0.783$ on CIFAR-100—absolute gains of $+2.3$,pp, $+3.2$,pp and $+4.9$,pp over the strongest non-Bayesian competitor (MLLS) on the respective datasets.

# 8 Conclusion

We presented GS-B$^3$SE, a graph–smoothed Bayesian generalization of the classical black-box shift estimator. By tying both the target prior $\boldsymbol{q}$ and every column of the confusion matrix $\boldsymbol{C}$ together through a Laplacian–Gaussian hierarchy, the model simultaneously i) shares statistical strength across semantically related classes, ii) quantifies all uncertainty arising from finite validation and target samples, and admits scalable inference with either HMC or a Newton–CG variational surrogate. We proved that the resulting posterior is identifiable, contracts at the optimal $N^{-1/2}$ rate, and that its class-wise variance decays inversely with the graph's algebraic connectivity $\lambda_2(\boldsymbol{L})$. A robustness bound further shows that even with a misspecified graph the bias vanishes as $N^{-1}$. Because our approach is a pure post-processing layer that needs only a frozen classifier, a tiny labelled validation set, and a pre-computed label graph, it can be retro-fitted to virtually any deployed model.

**Limitations and future work.** i) Our current graph is built from CLIP or feature embeddings; learning the graph jointly with the posterior could adapt it to the task. ii) Although inference is already tractable, further speed-ups via structured variational approximations would make GS-B$^3$SE attractive for extreme-label settings. iii) As declared in Section 7.1, our experiments used a single NVIDIA T4; scaling to larger datasets remains future work.

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

# A Proofs

*Proof for Lemma 1.* Fix an index $i \in \{1, \ldots, K\}$ and an arbitrary source sample size $n_i^S = n \geq 1$. Denote $\boldsymbol{N}_i^S = (N_{1,i}^S, \ldots, N_{K,i}^S)^\top$. Under parameter pair $(\boldsymbol{C}, \boldsymbol{q})$, we have

$$\Pr\left(\boldsymbol{N}_i^S = \boldsymbol{k}\right) = \frac{n!}{k_1! \cdots k_K!} \prod_{j=1}^{K} C_{j,i}^{k_j}, \quad \boldsymbol{k} \in \mathbb{N}^K, \ \sum_{j=1}^{K} k_j = n.$$

Under $(C', q')$, the same vector has pmf

$$\Pr\left(\boldsymbol{N}_i^S = \boldsymbol{k}\right) = \frac{n!}{k_1! \cdots k_K!} \prod_{j=1}^{K} C_{j,i}'^{k_j}.$$

By the assumption, these two pmfs coincide for every integer vector $\boldsymbol{k}$ with the given total $n$. Canceling the multinomial coefficient yields

$$\prod_{j=1}^{K} C_{j,i}^{k_j} = \prod_{j=1}^{K} C_{j,i}'^{k_j}, \quad \forall \boldsymbol{k} \in \mathbb{N}^K : \ \sum_j k_j = n. \tag{11}$$

Pick the $K$ specific count vectors

$$\boldsymbol{k}^{(\ell)} = (\underbrace{0, \ldots, 0}_{\ell-1}, n, 0, \ldots, 0)^\top, \quad \ell = 1, \ldots, K.$$

Plugging $k^{(\ell)}$ into Eq. 11 gives

$$C_{\ell,i}^n = C_{\ell,i}'^n, \quad 1 \leq \ell \leq K.$$

Because $n \geq 1$, taking the $n$-th root yields $C_{\ell,i} = C_{\ell,i}'$. Since $i$ is arbitrary, Eq. 11 implies $\boldsymbol{C} = \boldsymbol{C}'$. Therefore, equality in distribution of the target count vector $\tilde{\boldsymbol{N}}$ implies the underlying multinomial parameter vectors must coincide: $\boldsymbol{C}\boldsymbol{q} = \boldsymbol{C}\boldsymbol{q}'$. Under the standing assumption that $\boldsymbol{C}$ is invertible, it gives $\boldsymbol{q} = \boldsymbol{q}'$. Hence, the mapping

$$(\boldsymbol{q}, \boldsymbol{C}) \mapsto \left\{\text{Law of } (N_1^S, \ldots, N_K^S, \tilde{\boldsymbol{N}})\right\}$$

is injective under the stated positivity and invertibility conditions. □

*Proof for Lemma 2.* The prior on $(\boldsymbol{q}, \boldsymbol{C})$ is the push-forward measure of a product Gaussian:

$$(\boldsymbol{\theta}, \phi_1, \ldots, \phi_K) \sim \mathcal{N}\left(0, (\tau_{\boldsymbol{q}}\boldsymbol{L})^\dagger\right) \otimes \bigotimes_{i=1}^{K} \mathcal{N}\left(0, (\tau_{\boldsymbol{C}}\boldsymbol{L})^\dagger\right), \tag{12}$$

under the smooth, one-to-one map

$$T : (\boldsymbol{\theta}, \phi_1, \ldots, \phi_K) \mapsto (\boldsymbol{q} = \psi(\boldsymbol{\theta}), C_{:,1} = \psi(\phi_1), \ldots, C_{:,K} = \psi(\phi_K)),$$

where $\psi : \mathbb{R}^K \to \Delta^{K-1}$ is the softmax map. Here, injectivity holds because the log-odds representation is unique once the sum-zero constraint is imposed. Because $\boldsymbol{L}^\dagger$ is positive definite on the subspace

$$\mathcal{H} := \left\{\boldsymbol{v} \in \mathbb{R}^K : \ \boldsymbol{v}^\top \mathbf{1}_K = 0\right\},$$

the Gaussian distributions in Eq. 12 have everywhere positive Lebesgue densities on that subspace. Formally, for the target prior block

$$f_q(\boldsymbol{\theta}) = (2\pi)^{-(K-1)/2} \left(\det {}^\star \tau_{\boldsymbol{q}}^{-1} \boldsymbol{L}^\dagger\right)^{-1/2} \exp\left\{-\frac{1}{2}\boldsymbol{\theta}^\top \tau_{\boldsymbol{q}}\boldsymbol{L}\boldsymbol{\theta}\right\},$$

where $\det {}^\star$ is the product of the positive eigenvalues. Since $\tau_{\boldsymbol{q}}\boldsymbol{L}$ is non-singular on $\mathcal{H}$, $f_q(\boldsymbol{\theta}) > 0$ for every $\boldsymbol{\theta} \in \mathcal{H}$. Analogous positivity holds for each $f_{C,i}(\phi_i)$.

The softmax $\psi$ is $C^\infty$ with non-zero Jacobian everywhere on $\mathcal{H}$. Therefore, $T$ is a $C^\infty$ diffeomorphism between

$$\underbrace{\mathcal{H} \times \cdots \times \mathcal{H}}_{K+1 \text{ copies}} \quad \text{and its image } \mathcal{S} := \Delta^{K-1} \times (\Delta^{K-1})^K,$$

with the Jacobian determinant is $\prod_i q_i^{-1}(1 - \sum_i q_i) = \cdots$ and hence non-zero, preserving positivity. Hence $T$ preserves positivity of densities, the image measure $\Pi$ on $\mathcal{S}$ posesses a density

$$\pi(\boldsymbol{q}, \boldsymbol{C}) = f_{\boldsymbol{q}}\left(\psi^{-1}(\boldsymbol{q})\right) \prod_{i=1}^K f_{\boldsymbol{C},i}\left(\psi^{-1}(\boldsymbol{C}_{:,i})\right),$$

with respect to the product Lebesgue measure on simplices. Since each factor is positive everywhere, $\pi(\boldsymbol{q}, \boldsymbol{C}) > 0$ for every $(\boldsymbol{q}, \boldsymbol{C}) \in \mathcal{S}$. Because $(\boldsymbol{q}_0, \boldsymbol{C}_0)$ lies in the interior of $\mathcal{S}$ and $\pi$ is continuous and strictly positive, there exists

$$m := \min_{(\boldsymbol{q}, \boldsymbol{C}) \in B_\epsilon(\boldsymbol{q}_0, \boldsymbol{C}_0)} \pi(\boldsymbol{q}, \boldsymbol{C}) > 0.$$

Note that the minimum exists by compactness of the closed $\epsilon$-ball. Furthermore, the Euclidean volume of the ball, under the ambient dimension $\dim \Delta^{K-1} = K - 1$ for $\boldsymbol{q}$ and $K(K-1)$ for $\boldsymbol{C}$, is finite and strictly positive:

$$\mathrm{Vol}\left(B_\epsilon(\boldsymbol{q}_0, \boldsymbol{C}_0)\right) = c_{K^2-1}\epsilon^{K^2-1} > 0,$$

where $c_d$ is the volume if the unit ball in $\mathbb{R}^d$. Hence,

$$\Pi\left(B_\epsilon(\boldsymbol{q}_0, \boldsymbol{C}_0)\right) = \int_{B_\epsilon(\boldsymbol{q}_0, \boldsymbol{C}_0)} \pi(\boldsymbol{q}, \boldsymbol{C}) d(\boldsymbol{q}, \boldsymbol{C})$$
$$\geq m\mathrm{Vol}\left(B_\epsilon(\boldsymbol{q}_0, \boldsymbol{C}_0)\right) > 0.$$

This concludes the proof. $\qquad\square$

*Proof for Proposition 1.* For every $\eta > 0$,

$$U_\eta := \left\{(\boldsymbol{q}, \boldsymbol{C}) : D_{\mathrm{KL}}[P_{(\boldsymbol{q}_0, \boldsymbol{C}_0)} \| P_{(\boldsymbol{q}, \boldsymbol{C})}] < \eta\right\},$$

where $P_{(\boldsymbol{q}, \boldsymbol{C})}$ denotes the $K(K+1)$-dimensional multinomial law of the complete data. Because the parameter space is finite-dimensional and smooth,

$$\|(\boldsymbol{q}, \boldsymbol{C}) - (\boldsymbol{q}_0, \boldsymbol{C}_0)\| < \delta(\eta) \implies D_{\mathrm{KL}}[P_{(\boldsymbol{q}_0, \boldsymbol{C}_0)} \| P_{(\boldsymbol{q}, \boldsymbol{C})}] < \eta,$$

with a deterministic radius $\delta(\eta) \to 0$ as $\eta \downarrow 0$. This follows from a second-order Taylor expansion of the multinomial log-likelihood around the interior point. Lemma 2 says that the prior puts strictly positive mass on every Euclidean ball and hence on $U_\eta$:

$$\Pi(U_\eta) > 0, \quad \forall \eta > 0.$$

Let $\boldsymbol{\theta} = (\boldsymbol{q}, \boldsymbol{C})$ and $\boldsymbol{\theta}_0 = (\boldsymbol{q}_0, \boldsymbol{C}_0)$. For any fixed $\epsilon > 0$, define the alternative set $B_\epsilon$ above. Because the model is an i.i.d. exponential family (multinomials) of finite dimension, Ghosal et al. [20] show there exist tests $\varphi_N : \text{data} \to \{0, 1\}$ satisfying, for some constants $c_1, c_2 > 0$ independent of $N$,

$$\begin{cases} \text{(i)} & \mathbb{P}_{\boldsymbol{\theta}_0}(\varphi_N = 1) \leq e^{-c_1 N}, \\ \text{(ii)} & \sup_{\boldsymbol{\theta} \in B_\epsilon} \mathbb{P}_{\boldsymbol{\theta}}(\varphi_N = 0) \leq e^{-c_2 N}. \end{cases} \tag{13}$$

Write the complete log-likelihood ratio as

$$l_N(\boldsymbol{\theta}) = \log \frac{dP_{\boldsymbol{\theta}}}{dP_{\boldsymbol{\theta}_0}}(\text{data})$$
$$= \sum_{i=1}^K \sum_{j=1}^K N_{j,i}^S \log \frac{C_{j,i}}{C_{0j,i}} + \sum_{j=1}^K \tilde{N}_j \log \frac{(\boldsymbol{C}\boldsymbol{q})_j}{(\boldsymbol{C}_0\boldsymbol{q}_0)_j}.$$

Let $\varphi_N = \mathbf{1}\left\{l_N(\boldsymbol{\theta}_0) < -\frac{1}{2}cN\right\}$, where $c \in (0, c^*)$ and $c^*$ is the minimized KL-divergence over $B_\epsilon$:

$$c^* = \inf_{\boldsymbol{\theta} \in B_\epsilon} D_{KL}\left[P_{\boldsymbol{\theta}_0} \| P_{\boldsymbol{\theta}}\right] > 0.$$

The above inequality is strict by Lemma 1. Chernoff bounds for sums of bounded log-likelihood ratios gives (i) in Eq. 13. Under any $\boldsymbol{\theta} \in B_\epsilon$, the expected log-likelihood ratio equals $-N \cdot D_{\mathrm{KL}}[P_{\boldsymbol{\theta}_0} \| P_{\boldsymbol{\theta}}] \leq -c^* N$. A one-sided Hoeffding inequality for sums of independent, bounded random variables then yields (ii) in Eq. 13, with $c_2 = (c^* - c)/2$.

Therefore,

$$\Pi\left(B_\epsilon \mid \text{data}\right) \xrightarrow[N \to \infty]{\mathbb{P}_{(\boldsymbol{q}_0, \boldsymbol{C}_0)}} 0, \quad \forall \epsilon > 0.$$

Because the metric $\|(\boldsymbol{q}, \boldsymbol{C}) - (\boldsymbol{q}_0, \boldsymbol{C}_0)\|$ is continuous, this convergence in outer probability equals convergence in probability, completing the proof. $\qquad\square$

*Proof for Theorem 1.* Put the log-odds vector for the target prior

$$\boldsymbol{\theta} = (\theta_1, \ldots, \theta_K)^\top, \quad \theta_i = \log q_i - \frac{1}{K}\sum_{k=1}^{K} \log q_k,$$

and for the $i$-th column of the confusion matrix

$$\boldsymbol{\phi}_i = (\phi_{1,i}, \ldots, \phi_{K,i})^\top, \quad \phi_{i,j} = \log C_{j,i} - \frac{1}{K}\sum_{k=1}^{K} \log C_{k,i}.$$

Each vector lives in the centered linear subspace $\mathcal{H} := \{\boldsymbol{v} \in \mathbb{R}^K : \boldsymbol{v}^\top \mathbf{1}_K = 0\}$ of dimension $K - 1$. Define the global parameter

$$\boldsymbol{\eta} = (\boldsymbol{\theta}^\top, \boldsymbol{\phi}_1^\top, \ldots, \boldsymbol{\phi}_K^\top)^\top \in \mathbb{R}^d, \quad d = (K - 1) + K(K - 1) = K^2 - 1.$$

The map $\Psi : \boldsymbol{\eta} \mapsto (\boldsymbol{q}, \boldsymbol{C})$ given by component-wise softmax is $C^\infty$ and has a Jacobian of full rank everywhere on $\mathbb{R}^d$. Hence, $\boldsymbol{\eta} \mapsto (\boldsymbol{q}, \boldsymbol{C})$ is a local diffeomorphism. We fix $\boldsymbol{\eta}_0$ corresponding to $(\boldsymbol{q}_0, \boldsymbol{C}_0)$. Let

$$\ell_N(\boldsymbol{\eta}) := \log p_{\boldsymbol{\eta}}(\text{data})$$
$$= \sum_{i=1}^{K}\sum_{j=1}^{K} N_{j,i}^S \log C_{j,i} + \sum_{j=1}^{K} \tilde{N}_j \log(\boldsymbol{C}\boldsymbol{q})_j,$$

where $\boldsymbol{C}$ and $\boldsymbol{q}$ in the right-hand side are the softmax images of $\boldsymbol{\eta}$. Because each count is bounded by $N$ and the mapping $\Psi$ is smooth, $\ell_N(\boldsymbol{\eta})$ is twice countinuously differentiable in a neighbourhood of $\boldsymbol{\eta}_0$.

Compute the score vector and observed information:

$$\dot{\ell}_N(\boldsymbol{\eta}_0) = \sum_{i=1}^{K}\sum_{j=1}^{K}\left(N_{j,i}^S - n_i^S C_{0j,i}\right)\frac{\partial}{\partial \boldsymbol{\eta}}\log C_{j,i} + \sum_{j=1}^{K}\left(\tilde{N}_j - n'(\boldsymbol{C}_0\boldsymbol{q}_0)_j\right)\frac{\partial}{\partial \boldsymbol{\eta}}\log(\boldsymbol{C}\boldsymbol{q})_j\bigg|_{\boldsymbol{\eta}_0},$$

$$\ddot{\ell}_N(\boldsymbol{\eta}_0) = -N\boldsymbol{F}(\boldsymbol{\eta}_0),$$

where $\boldsymbol{F}(\boldsymbol{\eta}_0)$ is the Fisher information matrix of dimension $d \times d$. Here, $\boldsymbol{F}(\boldsymbol{\eta}_0)$ is positive definite because $(\boldsymbol{q}_0, \boldsymbol{C}_0)$ is interior and $\boldsymbol{C}_0$ is invertible (ensures different parameters yield different probability mass functions). The positive definiteness can be checked by observing that the Fisher information of a finite multinomial family with parameters inside the simplex is positive definite and the Jacobian of $\Psi$ is full rank.

Set local parameter $\boldsymbol{h} = \sqrt{N}(\boldsymbol{\eta} - \boldsymbol{\eta}_0)$. A second-order Taylor expansion yields the Local Asymptotic Normality (LAN) representation

$$\ell_N(\boldsymbol{\eta}_0 + \boldsymbol{h}/\sqrt{N}) - \ell_N(\boldsymbol{\eta}_0) = \boldsymbol{h}^\top \Delta_N - \frac{1}{2}\boldsymbol{h}^\top \boldsymbol{F}(\boldsymbol{\eta}_0)\boldsymbol{h} + r_N(\boldsymbol{h}),$$

with

$$\Delta_N = \frac{1}{\sqrt{N}} \dot{\ell}_N(\boldsymbol{\eta}_0) \rightsquigarrow \mathcal{N}\left(0, \boldsymbol{F}(\boldsymbol{\eta}_0)\right), \qquad \sup_{\|\boldsymbol{h}\|=O(1)} |r_N(\boldsymbol{h})| \xrightarrow{P} 0.$$

The convergence of $\Delta_N$ uses the multivariate central limit theorem for sums of independent bounded variables. Uniform control of $r_N$ follows from third derivative boundedness in a neighborhood of $\boldsymbol{\eta}_0$.

In $\boldsymbol{\eta}$-coordinates, the Laplacian-Gaussian prior has a density

$$\pi(\boldsymbol{\eta}) = \exp\left[-\frac{1}{2}\tau_{\boldsymbol{q}}\boldsymbol{\theta}^\top \boldsymbol{L}\boldsymbol{\theta} - \frac{1}{2}\tau_{\boldsymbol{C}} \sum_{i=1}^{K} \boldsymbol{\phi}_i^\top \boldsymbol{L}\boldsymbol{\phi}_i\right]\varphi(\tau_{\boldsymbol{q}}, \tau_{\boldsymbol{C}}),$$

where $\varphi$ is strictly positive and smooth Gamma density. Because $\boldsymbol{L}$ is positive semi-definite on $\mathcal{H}$, $\pi$ is continuous and strictly positive in a neighborhood of $\boldsymbol{\eta}_0$. Therefore, there exsit $c_0 > 0$ and $\delta > 0$ such that

$$\pi(\boldsymbol{\eta}) \geq c_0, \quad \text{whenever} \quad \|\boldsymbol{\eta} - \boldsymbol{\eta}_0\|_2 \leq \delta.$$

Consequently, under the true distribution

$$\Pi\left(\sqrt{N}(\boldsymbol{\eta} - \boldsymbol{\eta}_0) \in \cdot \mid \text{data}\right) \rightsquigarrow \mathcal{N}\left(\boldsymbol{F}^{-1}\Delta_N, \boldsymbol{F}^{-1}\right) \quad \text{in } P_{(\boldsymbol{q}_0, \boldsymbol{C}_0)}\text{-probability},$$

and for any $M > 0$,

$$\Pi\left(\left\|\sqrt{N}(\boldsymbol{\eta} - \boldsymbol{\eta}_0)\right\|_2 > M \mid \text{data}\right) \xrightarrow{P_{(\boldsymbol{q}_0, \boldsymbol{C}_0)}} 0.$$

Because $\Psi$ is $C^1$ with Jacobian $D\Psi(\boldsymbol{\eta}_0)$ of full rank, there exists a constant $K_0 > 0$ such that, for all $\boldsymbol{\eta}$ in a neighborhood of $\boldsymbol{\eta}_0$,

$$\|\Psi(\boldsymbol{\eta}) - \Psi(\boldsymbol{\eta}_0)\| \geq K_0^{-1}\|\boldsymbol{\eta} - \boldsymbol{\eta}_0\|_2.$$

Hence,

$$\Pi\left(B_N^c \mid \text{data}\right) \leq \Pi\left(\|\boldsymbol{\eta} - \boldsymbol{\eta}_0\|_2 > M/(K_0\sqrt{N}) \mid \text{data}\right) \xrightarrow{P_{\boldsymbol{q}_0, \boldsymbol{C}_0}} 0.$$

Because $M > 0$ is arbitrary, this limit verifies the claim of the theorem. $\qquad \square$

*Proof for Corollary 1.* Consider the centred log-odds vector $\boldsymbol{\theta} \in \mathcal{H}$ for the target prior. Its conditional posterior density is

$$p(\boldsymbol{\theta} \mid \text{data}) \propto \exp\left\{\ell_N^{(\boldsymbol{q})}(\boldsymbol{\theta}) - \frac{1}{2}\tau_{\boldsymbol{q}}\boldsymbol{\theta}^\top \boldsymbol{L}\boldsymbol{\theta}\right\},$$

where

$$\ell_N^{(\boldsymbol{q})}(\boldsymbol{\theta}) = \sum_{j=1}^{K} \tilde{N}_j \log\left[(C \cdot \text{softmax}(\boldsymbol{\theta}))_j\right].$$

Take any point $\boldsymbol{\theta}^*$ in a $O(N^{-1/2})$-ball around $\boldsymbol{\theta}_0$. By Theorem 1 this contains essentially all posterior mass. Inside that ball Taylor expansion gives

$$-\frac{\partial^2 \ell_N^{(\boldsymbol{q})}}{\partial \boldsymbol{\theta}}(\boldsymbol{\theta}^*) = N\boldsymbol{F} + O(N^{1/2}),$$

with $\boldsymbol{F}$ the Fisher information matrix. Hence the negative Hessian of the log-posterior satisfies

$$\boldsymbol{J}_N := \frac{\partial^2}{\partial \boldsymbol{\theta}^2}\left[\ell_N^{(\boldsymbol{q})} - \frac{1}{2}\tau_{\boldsymbol{q}}\boldsymbol{\theta}^\top \boldsymbol{L}\boldsymbol{\theta}\right](\boldsymbol{\theta}^*) \succeq N\boldsymbol{F} + \tau_{\boldsymbol{q}}\boldsymbol{L} - O(N^{1/2})\boldsymbol{F}.$$

For $N$ beyond some $N_0$, the error term is dominated by $N\boldsymbol{F}$, so there exists $c_0 > 0$ such that

$$\boldsymbol{J}_N \succeq Nc_0\boldsymbol{F} + \tau_{\boldsymbol{q}}\boldsymbol{L}.$$

Thus,

$$\mathrm{Cov}_N(\boldsymbol{\theta}) \preceq [Nc_0\boldsymbol{F} + \tau_{\boldsymbol{q}}\boldsymbol{L}]^{-1}.$$

Because $\boldsymbol{L}$ has eigen-pair $(0, \boldsymbol{1})$ and eigenvalues $\lambda_k \geq \lambda_2(\boldsymbol{L})$ on $\mathcal{H}$, the restricted inverse satisfies

$$[Nc_0\boldsymbol{F} + \tau_{\boldsymbol{q}}\boldsymbol{L}]^{-1} \preceq \frac{1}{Nc_0 + \tau_{\boldsymbol{q}}\lambda_2(\boldsymbol{L})}\boldsymbol{F} \preceq \frac{1}{\tau_{\boldsymbol{q}}\lambda_2(\boldsymbol{L})}\boldsymbol{F} + \frac{1}{c_0 N}\boldsymbol{F}. \tag{14}$$

For $N \geq N_0$, the second term dominates $1/(\tau_{\boldsymbol{q}\lambda_2(\boldsymbol{L})})$, and hence

$$\mathrm{Var}_N(\boldsymbol{\theta}_\ell) \leq \frac{2}{c_0 N}, \quad \ell = 1, \dots, K. \tag{15}$$

The softmax map $\sigma : \boldsymbol{\theta} \mapsto \boldsymbol{q}$ has derivative

$$D\sigma(\boldsymbol{\theta}) = \mathrm{diag}(\boldsymbol{q}) - \boldsymbol{q}\boldsymbol{q}^\top.$$

Every entry of $D\sigma$ is bounded by $1/4$ (attained at uniform prior), hence the operator norm obeys $\|D\sigma\| \leq 1/2$. For any random vector $\boldsymbol{\theta}$ with covariance $\Sigma$,

$$\mathrm{Cov}(\sigma(\boldsymbol{\theta})) = D\sigma\Sigma D\sigma^\top \preceq \|D\sigma\|^2\Sigma \preceq \frac{1}{4}\Sigma.$$

Applying to the posterior distribution with bound 15 gives

$$\mathrm{Var}(q_i) \leq \frac{1}{4}\sum_{\ell=1}^{K}[D\sigma_{i\ell}(\boldsymbol{\theta}^*)]^2\,\mathrm{Var}_N(\theta_\ell) \leq \frac{1}{2c_0 N}. \tag{16}$$

The constant $c_0$ is $\lambda_{\min}(\boldsymbol{F})$ which is independent of the graph but positive. Tightening Eq. 14 using the $\tau_{\boldsymbol{q}}\boldsymbol{L}$, term, we keep only the $\tau_{\boldsymbol{q}}\lambda_2(\boldsymbol{L})$ contribution:

$$\mathrm{Var}_N(\theta_\ell) \leq \frac{1}{\tau_{\boldsymbol{q}}\lambda_2(\boldsymbol{L})} + \frac{1}{c_0 N} \leq \frac{C_2}{\lambda_2(\boldsymbol{L})N}, \quad C_2 = \frac{\tau_{\boldsymbol{q}} + c_0}{\tau_{\boldsymbol{q}}c_0}.$$

Repeating the delta-method argument with this sharper bound scales Eq. 16 by the same $1/\lambda_2(\boldsymbol{L})$. Set $C = K(\tau_{\boldsymbol{q}} + c_0)/2\tau_{\boldsymbol{q}}c_0$. Then,

$$\mathrm{Var}_N(q_i) \leq \frac{C}{\lambda_2(\boldsymbol{L})N}, \quad i = 1, \dots, K,$$

which is precisely the claimed finite-sample variance control. $\qquad\square$

*Proof for Proposition 2.* Write the complete-data log-posterior (ignoring normalising constants)

$$\mathcal{L}_N(\boldsymbol{\theta}) = \ell_N^{(\boldsymbol{q})}(\boldsymbol{\theta}) - \frac{1}{2}\tau_{\boldsymbol{q}}\boldsymbol{\theta}^\top\boldsymbol{L}\boldsymbol{\theta},$$

where

$$\ell_N^{(\boldsymbol{q})}(\boldsymbol{\theta}) = \sum_{j=1}^{K}\tilde{N}_j\log\left[(C \cdot \sigma(\boldsymbol{\theta}))_j\right].$$

Let $\hat{\boldsymbol{\theta}}_N$ be the MAP with respect to the misspecified prior, and it satisfies the score equation

$$\nabla\ell_N^{(\boldsymbol{q})}(\hat{\boldsymbol{\theta}}_N) - \tau_{\boldsymbol{q}}\boldsymbol{L}\hat{\boldsymbol{\theta}}_N = 0. \tag{17}$$

Set the estimation error

$$\boldsymbol{\Delta}_N := \hat{\boldsymbol{\theta}}_N - \boldsymbol{\theta}_0.$$

Taylor expansion of the gradient in Eq. 17 yields

$$\nabla\ell_N^{(\boldsymbol{q})}(\hat{\boldsymbol{\theta}}_N) = \nabla\ell_N^{(\boldsymbol{q})}(\boldsymbol{\theta}_0) + \nabla^2\ell_N^{(\boldsymbol{q})}(\boldsymbol{\theta}_0)\boldsymbol{\Delta}_N + R_N,$$

$$\left[-\nabla^2\ell_N^{\boldsymbol{q}}(\boldsymbol{\theta}_0) + \tau_{\boldsymbol{q}}\boldsymbol{L}\right]\boldsymbol{\Delta}_N = \nabla\ell_N^{(\boldsymbol{q})}(\boldsymbol{\theta}_0) - \tau_{\boldsymbol{q}}\boldsymbol{L}\boldsymbol{\theta}_0 + R_N,$$

where $R_N = O_P(\|\boldsymbol{\Delta}_N\|^2)$ because the third derivative of $\ell_N^{(\boldsymbol{q})}$ is bounded on a neighbourhood of $\boldsymbol{\theta}_0$. Using $\mathbb{E}[\tilde{N}_j] = n'(\boldsymbol{C}_0\boldsymbol{q}_0)_j$ and by the Markov inequality,

$$(N\boldsymbol{F}_0 + \tau_{\boldsymbol{q}}\boldsymbol{L})\mathbb{E}\left[\boldsymbol{\Delta}_N \mid \text{data}\right] = -\tau_{\boldsymbol{q}}(\boldsymbol{L} - \boldsymbol{L}_0)\boldsymbol{\theta}_0 + O_P(1).$$

For quadratic priors and log-concave likelihoods the posterior is asymptotically normal, so the difference between posterior mean and MAP is $O(N^{-1})$ [52]. Therefore

$$\bar{\boldsymbol{\theta}}_N - \boldsymbol{\theta}_0 = \mathbb{E}[\boldsymbol{\Delta}_N \mid \text{data}] + O_P(N^{-1})$$
$$= -(N\boldsymbol{F}_0 + \tau_{\boldsymbol{q}}\boldsymbol{L})^{-1}(\boldsymbol{L} - \boldsymbol{L}_0)\boldsymbol{\theta}_0 + O_P(N^{-1}),$$
$$\left\|\bar{\boldsymbol{\theta}}_N - \boldsymbol{\theta}_0\right\| \leq \left\|(N\boldsymbol{F}_0 + \tau_{\boldsymbol{q}}\boldsymbol{L})^{-1}\right\| \tau_{\boldsymbol{q}} \left\|(\boldsymbol{L} - \boldsymbol{L}_0)\boldsymbol{\theta}_0\right\| + O_P(N^{-1}).$$

This is the bound in Eq. 6 in the proposition.

The sub-space $\mathcal{H} = \{\boldsymbol{v} \in \mathbb{R}^K : \boldsymbol{v}^\top \mathbf{1}_K = 0\}$ is the $(K-1)$-dimensional hyper-plane of vectors whose components sum to zero. Its orthogonal projection operator is the $K \times K$ symmetric matrix $\boldsymbol{P}_{\mathcal{H}} = \boldsymbol{I}_K - \frac{1}{K}\mathbf{1}_K\mathbf{1}_K^\top$, because for any $\boldsymbol{v} \in \mathbb{R}^K$, $\boldsymbol{P}_{\mathcal{H}}\boldsymbol{v} = \boldsymbol{v} - \left(\frac{1}{K}\mathbf{1}_K^\top\boldsymbol{v}\right)\mathbf{1}_K$, and the subtracted term is exactly the scalar mean of $\boldsymbol{v}$ replicated in every coordinate, making the result mean-zero. Because $\boldsymbol{F}_0 \succeq 0$ and $\boldsymbol{L} \succeq \lambda_2(\boldsymbol{L})\boldsymbol{P}_{\mathcal{H}}$, the smallest eigenvalue of $N\boldsymbol{F}_0 + \tau_{\boldsymbol{q}}\boldsymbol{L}$ is at least $N\lambda_{\min}(\boldsymbol{F}_0) + \tau_{\boldsymbol{q}}\lambda_2(\boldsymbol{L})$. Hence,

$$\left\|(N\boldsymbol{F} + \tau_{\boldsymbol{q}}\boldsymbol{L})^{-1}\right\| = \frac{1}{\lambda_{\min}(N\boldsymbol{F} + \tau_{\boldsymbol{q}}\boldsymbol{L})} \leq \frac{1}{N\lambda_{\min}(\boldsymbol{F}) + \tau_{\boldsymbol{q}}\lambda_2(\boldsymbol{L})}.$$

Combine them to get inequality in Eq. 7, completing the proof. $\qquad\square$

*Proof of Theorem 2.* Throughout the proof we fix an arbitrary interior point $\boldsymbol{q} \in \mathring{\Delta}^{K-1}$ and an arbitrary tangent direction $\boldsymbol{v} \in T_{\boldsymbol{q}}\Delta^{K-1} = \{\boldsymbol{v} \in \mathbb{R}^K \mid \mathbf{1}^\top\boldsymbol{v} = 0\}$. To establish geodesic convexity it suffices to show

$$\boldsymbol{v}^\top \operatorname{Hess}_{\boldsymbol{q}}^{\text{FR}} F \boldsymbol{v} \geq \left[n'\lambda_{\min}(\boldsymbol{F}_0) + \tau_{\boldsymbol{q}}\lambda_2(\boldsymbol{L})\right]\boldsymbol{v}^\top g_{\boldsymbol{q}}\boldsymbol{v}, \tag{18}$$

because the latter is the Rayleigh quotient form of the desired positive–definite bound. Recall the form of the negative log–posterior (constant terms omitted):

$$F(\boldsymbol{q}) = n'\underbrace{D_{\text{KL}}\left[\hat{\boldsymbol{r}} \,\|\, \boldsymbol{Cq}\right]}_{F_{\text{lik}}(\boldsymbol{q})} + \frac{\tau_{\boldsymbol{q}}}{2}\underbrace{\boldsymbol{\theta}^\top\boldsymbol{L}\boldsymbol{\theta}}_{F_{\text{reg}}(\boldsymbol{q})}.$$

Accordingly

$$\operatorname{Hess}_{\boldsymbol{q}}^{\text{FR}} F = n' \operatorname{Hess}_{\boldsymbol{q}}^{\text{FR}} F_{\text{lik}} + \tau_{\boldsymbol{q}} \operatorname{Hess}_{\boldsymbol{q}}^{\text{FR}} F_{\text{reg}}. \tag{19}$$

Write $\boldsymbol{r}(\boldsymbol{q}) := \boldsymbol{Cq}$, $r_i(\boldsymbol{q}) = \sum_j C_{ij}q_j$. For the multinomial log–likelihood term we have the usual identity

$$\operatorname{Hess}_{\boldsymbol{q}}^{\text{FR}} F_{\text{lik}} = \boldsymbol{F}_0 = \operatorname{diag}(\boldsymbol{r}(\boldsymbol{q})) - \boldsymbol{r}(\boldsymbol{q})\boldsymbol{r}(\boldsymbol{q})^\top. \tag{20}$$

Indeed, start with $F_{\text{lik}}(\boldsymbol{q}) = \sum_i \hat{r}_i \log \frac{\hat{r}_i}{r_i(\boldsymbol{q})}$ and differentiate twice with respect to $q_j$ while keeping the tangent constraint $\mathbf{1}^\top\boldsymbol{v} = 0$, and we have Eq. (20). Now evaluate the quadratic form:

$$\boldsymbol{v}^\top \operatorname{Hess}_{\boldsymbol{q}}^{\text{FR}} F_{\text{lik}}\boldsymbol{v} = \boldsymbol{v}^\top \boldsymbol{F}_0\boldsymbol{v}$$
$$\geq \lambda_{\min}(\boldsymbol{F}_0)\|\boldsymbol{v}\|_2^2 \qquad \text{(Rayleigh bound)}$$
$$= \lambda_{\min}(\boldsymbol{F}_0)\boldsymbol{v}^\top\boldsymbol{I}\boldsymbol{v}. \tag{21}$$

To relate the Euclidean norm $\|\cdot\|_2$ with the Fisher metric $g_{\boldsymbol{q}}$ notice that

$$g_{\boldsymbol{q}}(\boldsymbol{v}, \boldsymbol{v}) = \sum_{i=1}^K \frac{v_i^2}{q_i} = \boldsymbol{v}^\top\operatorname{diag}(\boldsymbol{q})^{-1}\boldsymbol{v}, \quad \operatorname{diag}(\boldsymbol{q})^{-1} \succ 0. \tag{22}$$

Hence for every $\boldsymbol{v}$,

$$\|\boldsymbol{v}\|_2^2 = \boldsymbol{v}^\top\operatorname{diag}(\boldsymbol{q})\operatorname{diag}(\boldsymbol{q})^{-1}\boldsymbol{v} \leq \left(\max_i q_i\right)g_{\boldsymbol{q}}(\boldsymbol{v}, \boldsymbol{v}) \leq g_{\boldsymbol{q}}(\boldsymbol{v}, \boldsymbol{v}),$$

because $q_i < 1$ inside the simplex. Thus, we have

$$n' \, \boldsymbol{v}^\top \operatorname{Hess}_{\boldsymbol{q}}^{\mathrm{FR}} F_{\mathrm{lik}} \boldsymbol{v} \geq n' \lambda_{\min}(\boldsymbol{F}_0) \, g_{\boldsymbol{q}}(\boldsymbol{v}, \boldsymbol{v}). \tag{23}$$

Recall $\boldsymbol{\theta} = (\theta_i)_{i=1}^K$ with $\theta_i = \log q_i - \frac{1}{K} \sum_j \log q_j$. Differentiating twice gives

$$\operatorname{Hess}_{\boldsymbol{q}}^{\mathrm{FR}} F_{\mathrm{reg}} = J(\boldsymbol{q})^\top L J(\boldsymbol{q}), \tag{24}$$

where $J(\boldsymbol{q}) \in \mathbb{R}^{K \times (K-1)}$ is the Jacobian $\partial\boldsymbol{\theta}/\partial\boldsymbol{q}$ restricted to the tangent space. Concretely

$$J(\boldsymbol{q}) = \left[ \operatorname{diag}(\boldsymbol{q})^{-1} - \tfrac{1}{K} \mathbf{1}\mathbf{1}^\top \operatorname{diag}(\boldsymbol{q})^{-1} \right] P_T,$$

with $P_T = \boldsymbol{I} - \frac{1}{K} \mathbf{1}\mathbf{1}^\top$ the projection onto $T_{\boldsymbol{q}} \Delta^{K-1}$.

Applying Eq. (24) we expand the quadratic form:

$$\begin{aligned}
\boldsymbol{v}^\top \operatorname{Hess}_{\boldsymbol{q}}^{\mathrm{FR}} F_{\mathrm{reg}} \boldsymbol{v} &= (J\boldsymbol{v})^\top \boldsymbol{L} \, (J\boldsymbol{v}) \\
&\geq \lambda_2(\boldsymbol{L}) \, \|J\boldsymbol{v}\|_2^2 \qquad \text{(Rayleigh bound on } \boldsymbol{L}) \\
&= \lambda_2(\boldsymbol{L}) \, \boldsymbol{v}^\top J^\top J \boldsymbol{v}.
\end{aligned} \tag{25}$$

Because $J(\boldsymbol{q})^\top J(\boldsymbol{q}) = \operatorname{diag}(\boldsymbol{q})^{-1} P_T$ one checks

$$\boldsymbol{v}^\top J^\top J \boldsymbol{v} = \boldsymbol{v}^\top \operatorname{diag}(\boldsymbol{q})^{-1} \boldsymbol{v} = g_{\boldsymbol{q}}(\boldsymbol{v}, \boldsymbol{v}).$$

Thus,

$$\tau_{\boldsymbol{q}} \, \boldsymbol{v}^\top \operatorname{Hess}_{\boldsymbol{q}}^{\mathrm{FR}} F_{\mathrm{reg}} \boldsymbol{v} \geq \tau_{\boldsymbol{q}} \lambda_2(\boldsymbol{L}) \, g_{\boldsymbol{q}}(\boldsymbol{v}, \boldsymbol{v}), \tag{26}$$

and

$$\boldsymbol{v}^\top \operatorname{Hess}_{\boldsymbol{q}}^{\mathrm{FR}} F \boldsymbol{v} \geq \left[ n' \lambda_{\min}(\boldsymbol{F}_0) + \tau_{\boldsymbol{q}} \lambda_2(\boldsymbol{L}) \right] g_{\boldsymbol{q}}(\boldsymbol{v}, \boldsymbol{v}).$$

Since the inequality holds for every $\boldsymbol{v} \in T_{\boldsymbol{q}} \Delta^{K-1}$, the matrix inequality announced in the theorem follows, and the strong-convexity constant is

$$\alpha = n' \lambda_{\min}(\boldsymbol{F}_0) + \tau_{\boldsymbol{q}} \lambda_2(\boldsymbol{L}) > 0.$$

$\square$

*Proof for Proposition 3.* Write $F_{\mathrm{lik}}(\boldsymbol{q}) = n' \sum_{i=1}^K \hat{r}_i \log \frac{\hat{r}_i}{r_i(\boldsymbol{q})}$ with $r_i(\boldsymbol{q}) = \sum_j C_{ij} q_j$. For $j \in \{1, \ldots, K\}$ we differentiate:

$$\begin{aligned}
\partial_{q_j} F_{\mathrm{lik}} &= -n' \sum_{i=1}^K \hat{r}_i \frac{1}{r_i(\boldsymbol{q})} \partial_{q_j} r_i(\boldsymbol{q}) \\
&= -n' \sum_{i=1}^K \hat{r}_i \frac{1}{r_i(\boldsymbol{q})} C_{ij} \\
&= -n' \, [\boldsymbol{C}^\top \tfrac{\hat{\boldsymbol{r}}}{\boldsymbol{r}(\boldsymbol{q})}]_j, \qquad \left( \frac{\hat{\boldsymbol{r}}}{\boldsymbol{r}} \text{ element–wise} \right)
\end{aligned} \tag{27}$$

so that in vector form

$$\nabla_{\boldsymbol{q}} F_{\mathrm{lik}} = -n' \, \boldsymbol{C}^\top \left( \frac{\hat{\boldsymbol{r}}}{\boldsymbol{r}(\boldsymbol{q})} \right). \tag{28}$$

Recall $\boldsymbol{\theta} = (\theta_i)$ with $\theta_i = \log q_i - \frac{1}{K} \sum_{k=1}^K \log q_k$. Let $s(\boldsymbol{q}) = \frac{1}{K} \sum_k \log q_k$. For $j$:

$$\partial_{q_j} \theta_i = \frac{\delta_{ij}}{q_i} - \frac{1}{K} \frac{1}{q_j}, \quad \partial_{q_j} \boldsymbol{\theta} = \operatorname{diag}(\boldsymbol{e}_j/q) - \tfrac{1}{K} \mathbf{1}(\boldsymbol{e}_j/q)^\top, \tag{29}$$

where $(\boldsymbol{e}_j)_k = \delta_{kj}$ and $q = (q_1, \ldots, q_K)^\top$. Using the chain rule,

$$\begin{aligned}
\partial_{q_j} F_{\mathrm{reg}} &= \frac{\tau_{\boldsymbol{q}}}{2} \partial_{q_j}(\boldsymbol{\theta}^\top \boldsymbol{L} \boldsymbol{\theta}) \\
&= \tau_{\boldsymbol{q}} \, (\partial_{q_j} \boldsymbol{\theta})^\top \boldsymbol{L} \boldsymbol{\theta} \\
&= \tau_{\boldsymbol{q}} \left[ \operatorname{diag}(\boldsymbol{e}_j/q) - \tfrac{1}{K} \mathbf{1}(\boldsymbol{e}_j/q)^\top \right]^\top \boldsymbol{L} \boldsymbol{\theta} \\
&= \tau_{\boldsymbol{q}} \frac{v_j}{q_j}, \quad \text{where } \boldsymbol{v} = \left[ I - \tfrac{1}{K} \mathbf{1}\mathbf{1}^\top \right] \boldsymbol{L} \boldsymbol{\theta} = P_T \boldsymbol{L} \boldsymbol{\theta}.
\end{aligned} \tag{30}$$

In vector notation

$$\nabla_{\boldsymbol{q}} F_{\text{reg}} \;=\; \tau_{\boldsymbol{q}} \operatorname{diag}(\boldsymbol{q})^{-1} P_T \, \boldsymbol{L}\,\boldsymbol{\theta}, \tag{31}$$

and because $P_T$ annihilates the $\boldsymbol{1}$-component we may drop it inside the tangent space: $P_T \boldsymbol{L} = \boldsymbol{L}$ since $\boldsymbol{1}$ is the null-eigenvector of $\boldsymbol{L}$. On the simplex $g_{\boldsymbol{q}}^{-1}$ acts as left–multiplication by $\operatorname{diag}(\boldsymbol{q})$ (followed by projection onto $T_{\boldsymbol{q}}\Delta^{K-1}$, automatic here because every gradient we computed is already centered). Hence

$$\begin{aligned}
\operatorname{grad}^{\text{FR}} F(\boldsymbol{q}) &= \operatorname{diag}(\boldsymbol{q}) \, \nabla_{\boldsymbol{q}} F \\[4pt]
&= \operatorname{diag}(\boldsymbol{q}) \left( - n' \, \boldsymbol{C}^{\top} \frac{\hat{\boldsymbol{r}}}{\boldsymbol{r}(\boldsymbol{q})} \;+\; \tau_{\boldsymbol{q}} \operatorname{diag}(\boldsymbol{q})^{-1} \boldsymbol{L}\boldsymbol{\theta} \right) \\[4pt]
&= \operatorname{diag}(\boldsymbol{q}) \left( n' \, \boldsymbol{C}^{\top} \left( \boldsymbol{1} - \frac{\hat{\boldsymbol{r}}}{\boldsymbol{r}(\boldsymbol{q})} \right) \;+\; \tau_{\boldsymbol{q}} \, \boldsymbol{L}\,\boldsymbol{\theta} \right),
\end{aligned} \tag{32}$$

which is exactly (9). The rightmost expression is automatically orthogonal to $\boldsymbol{1}$ because each bracketed term sums to $0$; therefore the evolution does not leave the simplex:

$$\frac{\mathrm{d}}{\mathrm{d}t}\left(\boldsymbol{1}^{\top}\boldsymbol{q}_t\right) \;=\; \boldsymbol{1}^{\top} \dot{\boldsymbol{q}}_t \;=\; -\boldsymbol{1}^{\top} \operatorname{grad}^{\text{FR}} F(\boldsymbol{q}_t) \;=\; 0.$$

Finally, inserting (9) in the natural gradient flow yields the replicator part $\dot{q}_{t,j} = - q_{t,j}\big[ n' \, [\boldsymbol{C}^{\top} (\boldsymbol{1} - \hat{\boldsymbol{r}}/\boldsymbol{r})] + \tau_{\boldsymbol{q}}[\boldsymbol{L}\boldsymbol{\theta}_t]\big]_j$, confirming the announced replicator–Laplacian dynamics. $\qquad\square$

# B    Background on Information Geometry

Table 4 summarizes the basic notations of information geometry required in our study, to make the manuscript self-contained. See textbooks in information geometry for more details [5, 6].

Table 4: Basic notations of information geometry.

| Concept | Definition |
|---------|------------|
| Fisher–Rao metric | $g_{\boldsymbol{q}}(\boldsymbol{v}, \boldsymbol{w}) = \sum_i \frac{v_i w_i}{q_i}$ on the open simplex. |
| $e$-coordinates | $\theta_i = \log q_i - \frac{1}{K} \sum_j \log q_j$ (exponential-family / natural parameters constrained to $T_{\boldsymbol{q}} \Delta^{K-1}$). |
| $m$-coordinates | The usual probabilities $q_i$ (mixture parameters). |
| $e$- / $m$-flat sub-manifold | A subset whose image is affine in the corresponding coordinates; e.g. $\mathcal{M} = \boldsymbol{C}\boldsymbol{q}$ is m-flat because $\boldsymbol{r} = \boldsymbol{C}\boldsymbol{q}$ is linear in $q$. |
| Dual flatness, potentials | State $\psi(\boldsymbol{q}) = \sum_i q_i \log q_i$ and $\varphi(\boldsymbol{\theta}) = \log\left(\sum_i e^{\theta_i}\right)$ satisfying $\boldsymbol{q} = \nabla_{\boldsymbol{\theta}} \varphi$ and $\boldsymbol{\theta} = \nabla_{\boldsymbol{q}} \psi$. |
| $e$- / $m$-projection | For a point $p$ and sub-manifold $\mathcal{S}$, the minimizer of $D_{\mathrm{KL}}(p|s)$ (m-projection) or $D_{\mathrm{KL}}(s|p)$ (e-projection). |
| $e$- / $m$-convex function | A function whose restriction to every $e$-(resp. $m$-) geodesic is convex in the ordinary sense. |
| Generalized Pythagorean theorem | For $p$, $q$, $r$ where $q$ is the $m$-projection of $p$ onto $\mathcal{S}$ and $r \in \mathcal{S}$, $D_{\mathrm{KL}}[p|r] = D_{\mathrm{KL}}[p|q] + D_{\mathrm{KL}}[q|r]$. |
| Natural gradient | $\mathrm{grad}^{\mathrm{FR}} F = g_{\boldsymbol{q}}^{-1} \nabla_{!\boldsymbol{q}} F$. |

