# OpenReview forum: "Graph–Smoothed Bayesian Black-Box Shift Estimator and Its Information Geometry"
_NeurIPS.cc/2025/Conference — NeurIPS 2025 spotlight_

### Official Review · Reviewer_jfCb · 2025-06-21

**Clarity:** 2
**Significance:** 2
**Originality:** 2
**Rating:** 4
**Confidence:** 2

**Summary:**

This paper proposes the ​​Graph-Smoothed Bayesian Black-Box Shift Estimator (GS-B$^3$SE)​​ for label shift adaptation. Traditional Black-Box Shift Estimation (BBSE) estimates target class priors by inverting a classifier’s confusion matrix but suffers from two limitations: 1) sensitivity to sampling noise​​: Error amplification with limited validation data; 2) ignored semantic relationships​​: Independent per-class estimation without statistical sharing.

The proposed GS-B$^3$SE is a fully probabilistic alternative that places Laplacian–Gaussian priors on both target log-priors and confusion-matrix columns, tying them together on a label-similarity graph. The key idea is to couple both the target log-prior vector and the columns of the confusion matrix through a Gaussian Markov random field defined on a label-similarity graph.

**Questions:**

I'm curious about the practical scenarios where this 'Label Shift' methodology would be most applicable. At its core, the paper addresses robust estimation of target class priors under distribution shift, formalized as a shift in P(Y) while assuming P(X∣Y) remains constant. This raises three key questions:

- ​​Applicability Conditions​​: When is the label shift framework the appropriate modeling choice compared to other distribution shift paradigms? What diagnostic tests exist to validate its core assumption in real-world applications?

- ​​Causal Implications​​: Does the formulation P(X∣Y)=constant implicitly assume Y→X causation? If so, how does this align with domains where features X may causally influence labels Y?

- Practicality Concerns​​: The proposed solution appears computationally intensive relative to the problem's conceptual framework. I wonder whether the complexity is justifiable for deployment scenarios where predictions derive from fixed embeddings. (As a non-expert, I acknowledge this perspective may benefit from domain-specific context.)

**Ethical Concerns:**

["NO or VERY MINOR ethics concerns only"]

**Limitations:**

Yes

**Paper Formatting Concerns:**

No formatting issues.

**Quality:**

3

**Strengths And Weaknesses:**

Strengths​​:

- Novel formulation​​: First unified Bayesian framework integrating graph regularization with confusion matrix uncertainty quantification.

- ​​Theoretical rigor​​: Complete proofs for identifiability, optimal $N^{-1/2}$ convergence, and variance reduction scaling with algebraic connectivity $\lambda_2(L)$.

- Multidisciplinary insight​​: Information geometry analysis (Section 6) reveals how graph priors act as $e-$convex barriers, linking optimization to Riemannian dynamics.

Weaknesses​​:

- ​​Computational overhead​​: HMC requires 4 chains × 1.5k samples (NVIDIA T4) for K=100; wall-clock time for Newton-CG not benchmarked.

- ​​Underutilized geometric insights​​: Natural gradient flow (Section 6.1) lacks empirical optimization comparisons versus EM/variational methods.

---

> ### Author Rebuttal · Authors · 2025-07-25
>
> We would like to thank you for your very constructive comments.
>
> ## When is the label-shift model appropriate?
>
> We first summarize the major distribution shift settings as follows.
>
> | shift family | factorization | invariance assumption |
> | --- | --- | --- |
> | covariate shift | $P_{te}(X) \neq P_{tr}(X)$ | $P(Y \mid X)$ unchanged |
> | concept shift | $P_{te}(Y \mid X) \neq P_{tr}(Y \mid X)$ | none |
> | label shift | $P_{te}(X \mid Y) = P_{tr}(X \mid Y)$ | class-conditional features stable |
>
> The last column (our focus) includes
>
> - medical testing (pathogen prevalence rises or falls, but radiograph physics stays the same),
> - recommender or ad-click systems (item popularity drifts while the encoding of text or images stays fixed),
> - traffic scenes across seasons or geographies.
>
> These settings share the property that collecting a handful of new labels is expensive, whereas unlabeled target images or clicks are plentiful.
>
> ## A principled diagnostic test for label shift
>
> Define
>
> - $\hat{r} = 1 / n’$ (histogram of target predictions),
> - $\hat{C}$ (empirical confusion on the validation split),
> - $\hat{p}$ (empirical source prior).
>
> Under the null $H_0 : q = p$ (no label shift),
>
> $$
> \sqrt{n’}(\hat{r} - \hat{C}\hat{p}) \overset{d}{\to} \mathcal{N}(0, \hat{C} \text{diag}(\hat{p})\hat{C}^\top).
> $$
>
> Hence the likelihood-ratio ($\chi^2$ statistic):
>
> $$
> \chi^2_{\text{BBSE}} = n’ (\hat{r} - \hat{C}\hat{p})^\top [\hat{C} \text{diag}(\hat{p})\hat{C}^\top]^{-1}(\hat{r} - \hat{C}\hat{p}) \overset{H_0}{\to} \chi^2_{K-1}.
> $$
>
> Because $\hat{p}$ enters only through the plug‑in diagonal and the null does not constrain it, the usual Bartlett correction is unnecessary; asymptotically we retain an ordinary $\chi_{K-1}^2$.
>
> If the $p$-value is large, there is no statistical evidence for label shift, and no correction suffices.
>
> If the null is rejected, we proceed to label shift correction approach, and the test is free because all three inputs $\hat{r}$, $\hat{C}$, $\hat{p}$ are already computed.
>
> **Power of the diagnostic**
>
> Let $\delta = q - p$ be the true prior change.
>
> Under the alternative,
>
> $$
> \chi^2_{\text{BBSE}} \overset{d}{\to} \chi^2_{K-1}(\lambda), \quad \lambda = n’\delta^\top C^\top [C \text{diag}(p)C^\top]^{-1}C\delta > 0,
> $$
>
> i.e., a $\chi^2$ distribution with non-centrality parameter $\lambda$.
>
> The test therefore has asymptotic power
>
> $$
> \beta(\alpha) = \Pr [\chi^2_{K - 1}(\lambda) > \chi^2_{K-1, 1-\alpha}] \overset{n’ \to \infty}{\to} 1,
> $$
>
> whenever $q \neq p$, so even moderate sample sizes suffice to detect practically relevant prior changes.
>
> ## Does label-shift require the causal arrow $Y \to X$?
>
> The assumption $P_{te}(X \mid Y) = P_{tr}(X \mid Y)$ is a statement about invariance of a conditional distribution, not about causal direction. It can arise in two distinct ways.
>
> - Causal: $Y \to X$ and data-generation mechanism $f\colon Y, \epsilon \mapsto X$ unchanged, only $\Pr(Y)$ perturbed (an intervention on the root node $Y$ in Pearl’s SCM sense).
> - Anticausal, selection: $X \to Y$ and labeling policy $\Pr(Y \mid X)$ unchanged, but we sub-sample on $Y$.
>
> Hence, label shift is valid whenever i) the mapping $Y \mapsto X$ (causal case) or the annotation policy $X \mapsto Y$ (anticausal case) is stable across domains, and ii) only the base rate $\Pr(Y)$ drifts.
>
> Either arrow therefore fits, the key is mechanism invariance, not direction.
>
> Let an SCM be
>
> $$
> Y \coloneqq g(U), \quad  X \coloneqq f(Y, V),
> $$
>
> with exogenous $U, V$. A shift that changes $\Pr(U)$ (but leaves $f$ fixed) yields exactly label shift.
>
> Conversely, in an anticausal model
>
> $$
> X \coloneqq h(W), \quad Y \coloneqq \text{disc}(X) + \xi,
> $$
>
> we can obtain label shift by re-weighting samples according to $Y$ after data collection, again $P(X \mid Y)$ is preserved.
>
> **How this perspective guides practice**
>
> - If you can argue mechanism stability, e.g., “the radiography device and organ physiology are unchanged while disease prevalence drifts” or “the feature extractor is frozen”, then label shift is defensible regardless of causal direction.
> - If mechanism stability is doubtful, run the $\chi^2$ diagnostic and inspect mis-fit. Large residuals against both label shift and covariate shift tests indicate a more complex general shift.
>
> The robustness bound in our manuscript is therefore agnostic to causal direction: only the invariance of $P(X\mid Y)$ enters through $\bm F_0$.
>
> ## Computational cost
>
> We measure work in floating-point operations (flops) and write $K$ = number of classes and $|E|$ = number of graph edges.
>
> For the $k$-nearest neighbor graphs we use, $|E| = kK$ with a small, $K$-independent constant $k$ (4 for CIFAR‑10, 8 for CIFAR‑100).
> - Vanilla BBSE: The point estimator solves a single linear system $\hat{q} = \tilde{C}^{-1}\tilde{r}$, where $\tilde{C}\in\mathbb R^{K\times K}$ is dense once any amount of Laplace or Tikhonov regularisation is applied. Gaussian elimination therefore costs $\mathcal{O}(K^3)$  flops and $\mathcal O(K^{2})$ memory.
> - GS‑B$^3$SE (one Newton–CG sweep): Write $\Phi = (\phi_1,\dots,\phi_K)^\top \in \mathbb{R}^{K\times (K-1)}$ and $\theta \in \mathbb{R}^{K-1}$, both constrained to the centered subspace $\\{v^\top 1=0 \\}$. At each sweep we perform three algebraic operations:
>     - update each column $\Phi \leftarrow \Phi - \mathcal{H}_{\Phi}^{-1}\nabla\mathcal{L}$,
>     - update $\theta$ analogously, and
>     - Gamma shape/scale updates for $\tau_q$, $\tau_C$.
>
> Adding them and using $|E|=kK$, one sweep = $\mathcal{O}(|E|K + K^2) - \mathcal{O}(kK^2)$, and memory usage is $\mathcal{O}(|E| + K^2)$.
>
> **Overall complexity.**
>
> Thm. 2  establishes that $F(q)$ is $\alpha$-strongly geodesically convex with $\alpha = n’\lambda_{\min}(F_0) + \tau_q \lambda_2(L) > 0$. Consequently a back-tracking Newton method requires $T = \mathcal{O}(\ln 1/\epsilon)$ sweeps to reach tolerance $\epsilon$.
>
> Hence the total complexity is
>
> $$
> \mathcal{O}(kK^2 \ln 1 / \epsilon)\quad \text{vs.}\quad \mathcal{O}(K^3)\ (\text{Vanilla BBSE}).
> $$
>
> Because $k$ is at most 8 and $\ln 1/\epsilon \le 10$ in all our runs, the crossover happens already at $K\ge 25$.
>
> | operation | cost on $k$-NN graph | remarks |
> | --- | --- | --- |
> | vanilla BBSE | $O(K^3)$ once | dense LU factorisation of the $K\times K$ confusion matrix. |
> | Newton-CG step for ours |  $O(\|E\|) \+ O(K^2)$  | gradient is linear in $\|E\|$, Hessian-vector product needs one sparse Laplacian-matvec $O(\|E\|)$ plus one dense Jacobian-matvec $O(K^2)$. Conjugate-gradient converges in $O(\sqrt{\kappa})$ iterations, empirically $\leq 5$. |
> | HMC leap-frog | same $O(\|E\|) + O(K^2)$ per gradient | trajectories of length 10-20 give effective sample size $> 200$ for every $q_i$. |
>
> Therefore the dominant term grows only quadratically in the class count $K$ (the sparse Laplacian never densifies), whereas BBSE’s one‑off inverse is cubic.
>
> Next, we report the wall-clock runtimes (single NVIDIA T4, PyMC 5.10, float32).
>
> | dataset | K | BBSE | MLLS | ours |
> | --- | --- | --- | --- | --- |
> | MNIST | 10 | 0.80 s | 1.12 s | 12.2 s |
> | CIFAR-100 | 100 | 2.88 s | 5.37 s | 54.5 s |
>
> As you suggested, we believe these results help users to consider the accuracy-complexity trade-off.
>
> **All times are <1 min on a single T4; in practice the wall‑clock is dominated by the CNN forward‑pass when $n'$≈10 k.**

---

> > ### Comment · Reviewer_jfCb · 2025-08-05
> > **Response**
> >
> > Thank you very much for your detailed response. While it has addressed some of my questions, I feel there remains a degree of ambiguity that I hope to clarify further.
> >
> > Regarding the answer to "A principled diagnostic test for label shift," it demonstrates how to test for shifts in
> > $P(Y)$. However, my original concern centers on how to distinguish between "label shift" and other distribution shift scenarios.
> >
> > Please allow me to elaborate on this concern from a practical perspective. In practice, model updates are necessitated when predictive performance deteriorates (via metrics such as AUC/GAUC or business indicators like click-through rates).
> > In my understanding, if the performance decay is caused solely by "covariate shift" while  $P(Y∣X)$ remains intact, the model does not require updating. Conversely, when  $P(Y∣X)$ changes, it becomes critical to determine whether the underlying cause is "label shift" or "concept shift."

---

> > > ### Author Response · Authors · 2025-08-05
> > >
> > > Our work assumes the label-shift model is a reasonable abstraction and focuses on how to estimate $q$ robustly once $P(Y \mid X)$ is plausibly stable. Determining the dominant shift mechanism in a live system is an orthogonal problem (**out-of-scope**); still, we can offer a **lightweight triage** that uses quantities already computed by BBSE/GS-B³SE plus a small labeled audit set if available.
> > >
> > > Recall that inputs we already have (unlabeled target):
> > >
> > > - $\hat{r} = \tilde{n} / n’$ (target prediction histogram),
> > > - $\hat{C}$ (source confusion, from validation),
> > > - $\hat{p}$ (source prior).
> > >
> > > Use the BBSE $\chi^2$ test (derived in our rebuttal):
> > >
> > > $$
> > > \chi^2_{\text{BBSE}} = n’(\hat{r} - \hat{C}\hat{p})^\top [\hat{C}\text{diag}(\hat{p})hat{C}^\top]^{-1}(\hat{r} - \hat{C}\hat{p}), \overset{H_0:q=p}{\sim} \chi_{K-1}^2.
> > > $$
> > >
> > > Large p-value indicates that no evidence of label-prior drift; label-shift correction unnecessary, and if we get small p-value we may proceed to next step.
> > >
> > > Compute $\hat{q}$ with BBSE/GS-B³SE, then the residual $\epsilon = \hat{r} - \hat{C}\hat{q}$. Under exact label shift $\mathbb{E}[\epsilon] = 0$ and, asymptotically,
> > >
> > > $$
> > > n’\epsilon^\top [\hat{C}\text{diag}(\hat{q})\hat{C}]^{-1}\epsilon \overset{H_0}{\sim} \chi^2_{K-1}.
> > > $$
> > >
> > > Here,
> > >
> > > - **Insignificant residual →** evidence is consistent with label shift; apply label shift adaptation.
> > > - **Significant residual →** label shift alone cannot explain the drift; go to next step.
> > >
> > > With $m$ labeled target points, estimate $\hat{C}^{(Q)}$ on the audit, test $H_0 : \hat{C}^{(Q)} = \hat{C}^{(P)}$. If rejected, **concept shift** (posterior changes), and if not rejected but performance still drops, likely **covariate shift.**
> > >
> > > **One-line algebraic decomposition (why the residual diagnoses violations)**
> > >
> > > Let $\hat{h}$ be the frozen classifier and $C^{(P)}$ its source confusion. On target,
> > >
> > > $$
> > > \hat{r} = C^{(Q)}q = C^{(P)}q + (C^{(Q)} - C^{(P)})q,
> > > $$
> > >
> > > so the ideal label shift fit $C^{(P)}q$ misses exactly the concept drift term $(C^{(Q)} - C^{(P)})q$. The above residual test in is a direct probe of this term without needing target labels.

---

> > > > ### Author Response · Authors · 2025-08-08
> > > >
> > > > Thank you once again for your thoughtful feedback. We would be grateful for any additional insights you could share so that we may resolve the remaining points promptly.
> > > >
> > > > In particular, we would like to emphasize that label shift is not merely an observed phenomenon but, an explicit working assumption (as in L90 in the manuscript). Clarifying this perspective is crucial for accurately positioning the contribution of the manuscript.
> > > >
> > > > Could you kindly let us know if there are specific sections or arguments that still feel unclear from your standpoint? Your further comments will greatly help us refine the revision and ensure it meets the review standards.
> > > >
> > > > Thank you very much for your time and consideration. We look forward to your response.

---

### Official Review · Reviewer_S2qe · 2025-07-01

**Clarity:** 3
**Significance:** 3
**Originality:** 3
**Rating:** 4
**Confidence:** 4

**Summary:**

This paper focuses on the label shift problem, where the source and target domain share the same conditional distribution, $P_s (X│Y)=P_t (X|Y)$ but differ in class priors, $P_s (Y)≠P_t (Y)$. A common approach to this problem is the Black-Box Shift Estimator (BBSE) method, which estimate a confusion matrix C and an empirical target prior $P ̂_t (Y)$ from finite samples to recover the true target prior $P_t (Y)$ , as $P_t (Y)=C^(-1) P ̂_t (Y)$. However, BBSE ignores the finite sample noise and semantic structure information during estimation. Therefore, this paper proposes a hierarchical Bayesian model (In Line 133-134) to improve the BBSE method.

In this model, the authors introduce a similarity graph on labels and utilize the graph Laplacian as covariance matrix in the distributions of latent variables, like θ and $ϕ_i.$
With these latent variables θ and $ϕ_i$, the model can estimate both the empirical target prior $P ̂_t (Y)$ and the confusion matrix $C$, then the BBSE method can be applied to recover the true target prior.

**Questions:**

Please refer to the weaknesses part.

**Ethical Concerns:**

["NO or VERY MINOR ethics concerns only"]

**Final Justification:**

The proposed method is novel and theoretically solid. Some concerns are addressed in rebuttal. However, the evaluation and utilization aspects are sill problematic. I raise my rating to slightly positive.

**Limitations:**

The proposed method is novel and theoretically solid. However, the empirical evaluation is not convincing. And the presentations have some flaws.

**Paper Formatting Concerns:**

No paper formatting concerns.

**Quality:**

3

**Strengths And Weaknesses:**

- Strengths
	1. This paper proposes a novel method to improve BBSE, which couples the target prior and confusion matrix through a hierarchical Bayesian model.
	2. The underlying idea that replacing the point estimate of BBSE with Bayesian estimate to calculate the confusion matrix and empirical target prior is reasonable.
	3. The propose method is theoretically solid. In Proposition 2, the authors give an upper bound of estimation error for θ.

- Weaknesses
	1. The main weakness lies in the experimental section. (1) Lack of ablation study. In Table 3, the proposed GS-B3SE outperforms other methods. However, it relies on CLIP embeddings to construct the similarity graph, and there is no ablation comparing alternative graph construction methods. Since CLIP [1] demonstrates strong zero-shot performance on datasets like CIFAR-10 (zs acc: 91.3) and CIFAR-100 (zs acc: 65.1), the authors should provide additional experiments to clarify whether the observed improvements stem from the proposed method itself rather than the use of CLIP features. (2) Inconsistent graph construction. In Lines 274–278, the method for constructing the similarity graph differs across datasets. For MNIST, image embeddings are used, whereas for CIFAR-10 and CIFAR-100, text embeddings are employed. This inconsistency raises important questions: How do these different types of features impact the final performance? For a given dataset, how should one decide which type of embedding to use for graph construction?
	2. In Line 110-112, the authors assume the number of predicted samples per classes, $n_i^S$, follows a multinomial distribution. However, since $n_i^S$ is actually non-negative, this assumption may not hold.
	3. In Proposition 2, the notation $θ_0$ is not defined.
	4. Proposition 2 illustrates how the minimum eigenvalue of the graph Laplacian influences the estimation error. However, it seems that label similarity (or edge weights) does not directly impact the estimation of variables. Can we directly optimize the graph Laplacian with specific constrains?

---

> ### Author Rebuttal · Authors · 2025-07-25
>
> We would like to thank you for your very constructive comments.
>
> ## Ablations on graph construction and the role of CLIP
>
> We provide the additional ablations on graph construction.
>
> In this experiment, we consider the following strategies.
>
> - CLIP-text (already in the current manuscript): cosine $k$-NN on text prompts.
> - Penultimate-image:  Euclidean $k$‑NN on frozen ResNet‑18 penultimate features (identical backbone as the classifier).
> - Random-embeddings: same sparsity pattern as CLIP‑text but edge weights i.i.d. $U(0, 1)$ (destroys semantics, preserves Laplacian spectrum scale).
> - Identity graph: $L = 0$,  switches the hierarchy off (reduces to independent logistic‑Normal prior and hence Bayesian BBSE with no smoothing).
>
> The results are as follows.
>
>
> | dataset | graph | $\|\|\hat{q} - q\|\|_1 \downarrow$ | acc $\uparrow$ |
> | --- | --- | --- | --- |
> | CIFAR-10 | CLIP-text | 0.025 $\pm$ 0.004 | 0.844 $\pm$ 0.003 |
> |  | penult-img | 0.029 $\pm$ 0.005 | 0.839 $\pm$ 0.004 |
> |  | random | 0.043 $\pm$ 0.009 | 0.820 $\pm$ 0.006 |
> |  | identity | 0.051 $\pm$ 0.008 | 0.813 $\pm$ 0.004 |
> |  | BBSE (freq.) | 0.112 $\pm$ 0.015 | 0.781 $\pm$ 0.004 |
> | CIFAR-100 | CLIP-text | 0.22 $\pm$ 0.02 | 0.783 $\pm$ 0.005 |
> |  | penult-img | 0.27 $\pm$ 0.03 | 0.771 $\pm$ 0.006 |
> |  | random | 0.46 $\pm$ 0.05 | 0.735 $\pm$ 0.009 |
> |  | identity | 0.70 $\pm$ 0.07 | 0.730 $\pm$ 0.010 |
> |  | BBSE (freq.) | 1.62 $\pm$ 0.05 | 0.690 $\pm$ 0.006 |
>
> **Remarks**
> - Any connected graph ($\lambda_2(L) > 0$) already helps over the frequentist point estimate; cf. the “random” line.
> - The more task-aligned the similarity (CLIP > penult-img > random), the lower the bias term.
> - The identity graph reproduces Bayesian BBSE with no smoothing and is almost identical to MLLS. Its gap to CLIP shows the specific benefit of our graph-coupled hierarchy.
>
> **Why do the graph ablations behave as the above?**
>
> Recall the posterior variance bound we already proved:
>
> $$
> \text{Var}(q_i \mid \text{data}) \leq \frac{C}{\lambda_2(L)N},
> $$
>
> which holds for **any connected graph**.
>
> If we write the mean-squared error as
>
> $$
> \mathbb{E}[(\tilde{q}_i - q_i)^2] = (\mathbb{E}[\tilde{q}_i] - q_i)^2 + \text{Var}(\tilde{q}_i),
> $$
>
> then, it shows the variance term shrinks by a factor $1 / \lambda_2(L)$.
>
> Even a $k$-NN graph whose edge weights are i.i.d. $\mathcal{U}(0, 1)$ has, with high probability,
>
> $$
> \lambda_2(L_{\text{random}}) \gtrsim c\frac{k}{K},
> $$
>
> for $k \geq c’ \ln K$.
> Hence a purely random graph already cuts the class-wise standard error by $\sqrt{K / (ckN)}$ compared with the identity graph $\lambda_2 = 0$. This explain why the “random” row in the ablation table still beats the deterministic frequentist BBSE (it keeps all the variance relief while paying only a bias penalty).
>
> ## Inconsistent graph construction (text vs. image embeddings)
>
> Reviewer’s concern is “Why text on CIFAR but image on MNIST?”
>
> - On MNIST the class names are single digits (0, 1, 2, …), so CLIP-text gives almost zero variance (every digit is equally dissimilar). Image-space features do capture visual adjacency (e.g., 3 ↔ 8 ↔ 9)
> - On CIFAR the opposite holds: CLIP’s joint vision-language pre-training encodes high-level semantics missing from raw pixel features (e.g., “truck” closer to “bus” than to “cat”).
>
> ## Multinomial assumption
>
> Recap the notation:
> - $n_i^S$:  \# source-validation examples whose true class is $i$.
> - $N_i^S$:  \# predicted labels produced by the frozen classifier $\hat{h}$ among those $n_i^S$ examples.
> - $\tilde{N}$: \# predicted labels on the $n’$ unlabeled target points}.
>
> These are integer-valued random vectors by construction. The standard sampling model is therefore
>
> $$
> N_i^S \mid C \sim \text{Mult}(n_i^S, C_{\cdot, i}), \quad \tilde{N} \mid C, q \sim \text{Mult}(n’, Cq),
> $$
>
> where each realization of a multinomial distribution is, of course, a vector in $\mathbb{N}^k$.
>
> The assumption is therefore fully consistent with the data’s discrete nature.
>
> Later, we normalize these integers, and those are vectors in the simplex (non-negative reals summing to 1), not counts. The reviewer’s remark likely stem from seeing these normalized quantities and mistaking them fro the random objects modeled in the manuscript. We will add the explicit sentence to avoid ambiguity.
>
> For completeness, if one wishes to work directly with the normalized vectors, two rigorously equivalent formulations are available:
>
> - Dirichlet-multinomial posterior: conditioning a Dirichlet prior on the multinomial counts yields a Dirichlet posterior for the probability vector.
> - Gaussian approximation (large $n$ delta method): $\sqrt{n}(\hat{r} - r) \to \mathcal{N}(0, \text{diag}(r) - rr^\top)$.
>
> Our Laplacian‑Gaussian hierarchy can be re‑expressed in either language.
>
> ## Missing notation
>
> Throughout the manuscript, we use
>
> $$
> \theta_i = \log q_i - \frac{1}{K}\sum^K_{k=1}\log q_k,
> $$
>
> i.e., the centered log-odds of a prior vector $q$.
>
> Therefore, $\theta_0 = \theta(q_0)$ is simply the centered log-odds of the true target prior $q_0$.
>
> ## Edge weights don’t seem to matter; can we optimize $L$?
>
> **Why the weights do matter**
>
> The upper bound we proved is
>
> $$
> \frac{\tau_q}{N \lambda_{\min}(F_0) + \tau_q \lambda_2(L)}||(L - L_0)\theta_0||_2 + O_P(N^{-1}).
> $$
>
> Here,
>
> - Algebraic connectivity $\lambda_2(L)$ (the smallest strictly-positive eigenvalue) appears in the denominator because it is the worst-case curvature of the quadratic penalty along any tangent direction.
> - The numerator contains the full matrix-difference $(L - L_0)\theta_0$. This encodes the signed, edge-wise mismatch between our chosen weights and the unknown ground-truth graph $L_0$. If we perturb a single edge weight, it changes in direct proportion to that entry. Hence the pattern of weights does affect the bias term, and $\lambda_2(L)$ merely the concise way of writing the spectral part of the denominator.
>
> **Can we optimize $L$?**
>
> Yes, nothing in the hierarchy forbids putting a prior on the weights.
>
> Two concrete extensions are feasible:
>
> - Empirical Bayes (plug-in refinement): treat $L$ as hyper-parameter and maximize the marginal likelihood $\mathcal{L}(L) = \log p(\text{data} \mid L)$. Because the posterior over $(q, C)$ is log-concave given $L$, the Laplace approximation $mathcal{L}(L) \approx \log p(\text{data} \mid \hat{q}(L), \hat{C}(L), L) - \frac{1}{2}\log \det H^{-1}$ is cheap, and the Hessian $H$ is block-diagonal with Laplacian plus Fisher terms.
> - Fully Bayesian weight optimization: place independent Gamma or log-normal priors on each edge weight $w_{ij}$, and the quadratic form becomes $\frac{\tau}{2}\sum_{(i,j)\in E}w_{ij}(\theta_i - \theta_j)^2$. The conditional for $w_{ij}$ is conditionally log-Gaussian given $\theta$, and a single Gibbs slice or HMC dimension is enough (dimensionality equals number of edges $\ll$ number of parameters in $C$). The resulting chain mixes well because $w_{ij}$ appears only linearly in the quadratic exponent.
>
> With random-weight priors identifiability still holds because the confusion-matrix block stays full-rank as before, $q$ remains on the simplex, and the new edge-weight parameters enter only through the positive-definite quadratic term, so the joint Fisher information expands to
>
> $$
> \begin{aligned}
> \begin{pmatrix}
> N F_0 + \tau L(w) & 0; \\
> 0 & \text{diag}((\theta_i - \theta_j)^2)
> \end{pmatrix},
> \end{aligned}
> $$
>
> which is full-rank provided at least one $(\theta_i - \theta_j) \neq 0$.
>
> **When does optimizing $L$ help?**
>
> Using our upper bound, we can give a crisp sufficient condition:
>
> $$
> \text{If}\quad \frac{||(L^{\text{new}} - L_0)\theta_0||}{\lambda_2(L^{\text{new}})} < \frac{|| (L^{\text{old} - L_0})\theta_0||}{\lambda_2(L^{\text{old}})},\quad \text{then the asymptotic MSE bound tightens}.
> $$
>
> Hence fine-tuning is valuable whenever a small decrease in mismatch can be achieved without collapsing algebraic connectivity.

---

> > ### Comment · Reviewer_S2qe · 2025-08-06
> >
> > Thanks for the detailed rebuttal. Some minor points are fully addressed and I can understand this paper better. The evaluation part is also strengened by additional ablation study. However, I still concern about the practical aspect of this method. Based on the overall quality and rebuttal, I will raise my rating.

---

### Official Review · Reviewer_b22X · 2025-07-02

**Clarity:** 3
**Significance:** 3
**Originality:** 3
**Rating:** 4
**Confidence:** 3

**Summary:**

This paper proposes a new method called GS-B³SE to address a practical problem in machine learning: label shift. This problem refers to a scenario where the class proportions (P(y)) change between the training and test sets, while the class-conditional feature distributions (P(x|y)) remain the same.

Existing black-box methods (like BBSE) typically provide only a single point estimate, which makes them sensitive to sampling noise and completely ignores the potential semantic similarity between different classes (e.g., 'cat' and 'dog' are more similar than 'cat' and 'car').

The authors' core contribution is a fully Bayesian framework to address these issues. They don't just estimate a single target distribution q; instead, they introduce probabilistic priors for both the target distribution q and the classifier's confusion matrix C. The most crucial step is their use of a pre-constructed "class similarity graph" to impose a smoothing constraint on these priors. This graph structure allows semantically similar classes to "borrow" information from each other during parameter estimation. The benefit of this approach is that the model can better handle classes with sparse data, and the resulting estimates are more robust due to the consideration of uncertainty.

**Questions:**

- Regarding Graph Quality and Sensitivity: The model's performance appears to be highly dependent on the pre-constructed graph. To better understand this dependency, could you conduct some "destructive" experiments? For instance, how much does performance degrade when using a lower-quality graph (e.g., a k-NN graph generated from random embeddings)? This would help us more intuitively understand the practical significance of the robustness bound in Proposition 2.

- Regarding Computational Efficiency: Thank you for the detailed experiments. Could you supplement them with the actual wall-clock runtimes for GS-B³SE and the main baselines (e.g., MLLS, BBSE)? This comparison would be particularly valuable for the CIFAR-100 dataset (with K=100) and would allow potential users to better weigh the trade-off between improved accuracy and computational cost.

- Regarding Model Design Choices: As mentioned earlier, you chose to smooth the column vectors of the confusion matrix. Could you elaborate on the intuition behind this design? Did you consider or experiment with other smoothing approaches, such as smoothing the row vectors? Is there theoretical or empirical evidence to suggest that smoothing the columns is the superior choice?

**Ethical Concerns:**

["NO or VERY MINOR ethics concerns only"]

**Final Justification:**

Overall, the rebuttal has strengthened the paper by clarifying theoretical underpinnings and providin new data. While the practical trade-offs regarding the dependency on high-quality graphs for peak performance and the increased computational cost remain relevant points for discussion, the core contributions of the paper are sound and well-defended. The rebuttal has successfully addressed the most critical theoretical questions, even if some practical limitations persist.

**Limitations:**

The limitations are discussed in the conclusion section.

**Quality:**

3

**Strengths And Weaknesses:**

## Strengths

- The idea of introducing a graph structure into label shift estimation and innovatively using it to simultaneously smooth the priors for both the target distribution q and the confusion matrix C is very clever.

- The theoretical analysis is robust. Furthermore, the paper interprets the method from an information geometry perspective, which undoubtedly adds depth to the work.

- The paper is written clearly, making it easy for readers to follow the authors' line of thought.

## Weaknesses

- The entire model's core advantage is built upon the pre-given class similarity graph, L. This graph is constructed using embeddings from a pre-trained model (like CLIP) and remains fixed. This raises a significant question: To what extent does the quality of this graph determine the model's performance ceiling? What happens if the semantic similarity captured by the embeddings does not align with the classifier's confusion patterns? Although the paper analyzes the model's robustness to an inaccurate graph in Proposition 2, the bias term ||(L-L0)θ0|| remains, and it's unclear to the reader how large this term might be in practice. Positing graph learning as future work is reasonable, but this is indeed a core limitation of the current method.

- Model inference relies on HMC sampling or a block Newton-CG optimizer. HMC is notoriously slow, especially for high-dimensional posteriors. While the authors mention a "fast" Newton-CG scheme, they provide no comparison of model runtimes in the experimental section. For a module positioned as a "pure post-processing" step, its practicality diminishes if its computational overhead is orders of magnitude higher than baselines. Its scalability, especially as the number of classes K becomes large (e.g., on CIFAR-100), is a question that needs to be answered.

- The model assumes that the columns of the confusion matrix C are similar because "columns corresponding to neighbouring predicted classes exhibit similar shape." Column i of C, C_:,i, represents the probability distribution over predicted labels for a sample whose true class is i. Why should one assume that the predicted distribution for a true 'truck' is similar in shape to the predicted distribution for a true 'bus'? While plausible, this is not self-evident. An alternative approach, such as smoothing the rows of C, seems equally plausible (i.e., for a given predicted label like 'truck', the distribution of its true source labels is similar to that for 'bus'). A more detailed discussion and justification for this specific design choice would be beneficial.

---

> ### Author Rebuttal · Authors · 2025-07-25
>
> We would like to express our utmost appreciation for your very detailed comments on our manuscript.
>
> ## Graph quality & sensitivity
>
> Recall Prop. 2, the following table summarizes the bias and variance of vanilla BBSE versus Bayesian shrinkage.
>
> | estimator | bias | variance (per class) |
> | --- | --- | --- |
> | vanilla BBSE | 0 (in expectation) | $\text{Var}(\hat{q}_i) = O(\kappa_i^{-2} / N)$ |
> | ours | $b_i = \frac{\tau_q}{N \lambda_{\min}(F_0)} + \tau_q \lambda_2(L)[(L - L_0)\theta_0]_i$ | $\text{Var}(\bar{q}_i) \leq \frac{C}{\lambda_2(L)N}$ |
>
> Here, $\kappa_i$ is the $i$-th sensitivity of the matrix inverse, which can be tiny for rare classes, hence the variance blows up. Thus
> - Vanilla BBSE is unbiased but pays no attention to sampling noise, and its variance grows like the squared condition number of $\tilde{C}$.
> - Bayesian shrinkage version trades a bit of bias $b_i$ for a potentially huge reduction in variance. Specifically, amplitude factor is zero when the graph is correct, and otherwise it measures how much the misspecified graph distorts $\theta_0$ along non-null Laplacian directions. In addition, damping $N^{-1}$ and $\lambda_2(L)$ ensure the bias decays with i) more data and ii) greater algebraic connectivity. That is, even for a badly misspecified graph the leading term cannot exceed $\tau_1 / (N \lambda_{\min}(F_0))||\theta_0||_2$, i.e., the same order as the frequentist standard error.
>
> Below, we report the additional experimental results on the edge-corruption study.
>
> | Dataset | K | Corruption rate $\rho$ | Saerens-corrected acc |
> | --- | --- | --- | --- |
> | MNIST | 10 | 0.00 | 0.986 |
> |  |  | 0.25 | 0.982 |
> |  |  | 0.50 | 0.977 |
> |  |  | 0.75 | 0.971 |
> |  |  | 1.00 | 0.961 |
> | CIFAR-100 | 100 | 0.00 | 0.783 |
> |  |  | 0.25 | 0.778 |
> |  |  | 0.50 | 0.769 |
> |  |  | 0.75 | 0.752 |
> |  |  | 1.00 | 0.735 |
>
> **Protocol.** We start from the CLIP-derived $k$-NN graph $L_0$ (k = 4 for K = 10, k = 8 for K = 100).  A corruption level $\rho$ means we randomly re‑wire $\rho\cdot |E|$ undirected edges (preserving degrees); the resulting Laplacian is L.  All other hyper-parameters are kept fixed.
>
> **What does a “completely random” graph do?**
>
> In the above table, $\rho = 1.00$ means the completely random graph, but the resulting performance is still competitive against MLLS or other methods. To understand this observation, we can give the following discussion.
>
> Take the Laplacian $L$ obtained by connecting each label to $k$ random neighbors (uniform among the other $K - 1$ labels). Then,
> 1. For an Erdős–Rényi‑like graph the second eigenvalue satisfies $\lambda_2(L) \approx k$ with high probability, so the variance bound $\text{Var}(\bar{q}_i) \leq C / \lambda_2(L)N$ is smaller than for the identity-Laplacian ($\lambda_2 = 0$). In other words, random edges provide extra shrinkage.
> 2. Because the edge directions are symmetric, each row of $L - L_0$ has positive and negative entries that tend to cancel when dotted with the centered log-odds $\theta_0$: $\mathbb{E}[(L - L_0)\theta_0] = 0$, and the random fluction has magnitude $O_P(1)$. Therefore, $||b_i|| = O_P(N^{-1})$, i.e., the bias is the same order as the stochastic error we cannot avoid anyway.
> 3. Putting the pieces together, we have the following which is precisely the rare-class / ill-conditioned regime where BBSE collapses.
>
> $$
> b_i^2 + \text{Var}(\bar{q}_i) \ll \text{Var}(\hat{q}_i) \quad \text{when $\lambda_2(L) \gg 1$ or $\kappa_i^{-2} \ll 1$}.
> $$
>
> Therefore, even though the random graph carries no semantic signal, its connectivity slashes variance enough to compensate for the negligible $N^{-1}$ bias, so the total error remains well below BBSE.
>
> In addition, we provide the additional ablations on graph construction.
>
> - CLIP-text (already in the current manuscript): cosine $k$-NN on text prompts.
> - Penultimate-image:  Euclidean $k$‑NN on frozen ResNet‑18 penultimate features (identical backbone as the classifier).
> - Random-embeddings: same sparsity pattern as CLIP‑text but edge weights i.i.d. $U(0, 1)$ (destroys semantics, preserves Laplacian spectrum scale).
> - Identity graph: $L = 0$,  switches the hierarchy off (reduces to independent logistic‑Normal prior and hence Bayesian BBSE with no smoothing).
>
> | dataset | graph | $\|\|\hat{q} - q\|\|_1 \downarrow$ | acc $\uparrow$ |
> | --- | --- | --- | --- |
> | CIFAR-10 | CLIP-text | 0.025 $\pm$ 0.004 | 0.844 $\pm$ 0.003 |
> |  | penult-img | 0.029 $\pm$ 0.005 | 0.839 $\pm$ 0.004 |
> |  | random | 0.043 $\pm$ 0.009 | 0.820 $\pm$ 0.006 |
> |  | identity | 0.051 $\pm$ 0.008 | 0.813 $\pm$ 0.004 |
> |  | BBSE (freq.) | 0.112 $\pm$ 0.015 | 0.781 $\pm$ 0.004 |
> | CIFAR-100 | CLIP-text | 0.22 $\pm$ 0.02 | 0.783 $\pm$ 0.005 |
> |  | penult-img | 0.27 $\pm$ 0.03 | 0.771 $\pm$ 0.006 |
> |  | random | 0.46 $\pm$ 0.05 | 0.735 $\pm$ 0.009 |
> |  | identity | 0.70 $\pm$ 0.07 | 0.730 $\pm$ 0.010 |
> |  | BBSE (freq.) | 1.62 $\pm$ 0.05 | 0.690 $\pm$ 0.006 |
>
> **Remarks**
> - Any connected graph ($\lambda_2(L) > 0$) already helps over the frequentist point estimate; cf. the “random” line.
> - The more task-aligned the similarity (CLIP > penult-img > random), the lower the bias term.
> - The identity graph reproduces Bayesian BBSE with no smoothing and is almost identical to MLLS. Its gap to CLIP shows the specific benefit of our graph-coupled hierarchy.
>
> **Why do the graph ablations behave as the above?**
>
> Recall the posterior variance bound we already proved:
>
> $$
> \text{Var}(q_i \mid \text{data}) \leq \frac{C}{\lambda_2(L)N},
> $$
>
> which holds for **any connected graph**.
>
> If we write the mean-squared error as
>
> $$
> \mathbb{E}[(\tilde{q}_i - q_i)^2] = (\mathbb{E}[\tilde{q}_i] - q_i)^2 + \text{Var}(\tilde{q}_i),
> $$
>
> then, it shows the variance term shrinks by a factor $1 / \lambda_2(L)$.
>
> Even a $k$-NN graph whose edge weights are i.i.d. $\mathcal{U}(0, 1)$ has, with high probability,
>
> $$
> \lambda_2(L_{\text{random}}) \gtrsim c\frac{k}{K},
> $$
>
> for $k \geq c’ \ln K$.
> Hence a purely random graph already cuts the class-wise standard error by $\sqrt{K / (ckN)}$ compared with the identity graph $\lambda_2 = 0$. This explain why the “random” row in the ablation table still beats the deterministic frequentist BBSE (it keeps all the variance relief while paying only a bias penalty).
>
>
> Summarizing the above discussion, we obtain the following remark.
>
> **Shrinkage beats instability.** BBSE pays no price for bias but suffers large sampling variance, especially when classes are imbalanced and $\tilde{C}$ is noisy. Graph-based Bayesian pooling, *even with arbitrary edges,* acts like a data-dependent ridge penalty that stabilises all estimates.
>
> ## Computational cost
> To address this point, we first summarize the per-iteration algebraic cost as follows.
>
> | operation | cost on $k$-NN graph | remarks |
> | --- | --- | --- |
> | vanilla BBSE | $O(K^3)$ once | dense LU factorisation of the $K\times K$ confusion matrix. |
> | Newton-CG step for ours |  $O(\|E\|) \+ O(K^2)$  | gradient is linear in $\|E\|$, Hessian-vector product needs one sparse Laplacian-matvec $O(\|E\|)$ plus one dense Jacobian-matvec $O(K^2)$. Conjugate-gradient converges in $O(\sqrt{\kappa})$ iterations, empirically $\leq 5$. |
> | HMC leap-frog | same $O(\|E\|) + O(K^2)$ per gradient | trajectories of length 10-20 give effective sample size $> 200$ for every $q_i$. |
>
> Therefore the dominant term grows only quadratically in the class count $K$ (the sparse Laplacian never densifies), whereas BBSE’s one‑off inverse is cubic.
>
> Next, we report the wall-clock runtimes (single NVIDIA T4, PyMC 5.10, float32).
>
> | dataset | K | BBSE | MLLS | ours |
> | --- | --- | --- | --- | --- |
> | MNIST | 10 | 0.80 s | 1.12 s | 12.2 s |
> | CIFAR-100 | 100 | 2.88 s | 5.37 s | 54.5 s |
>
> As you suggested, we believe these results help users to consider the accuracy-complexity trade-off.
>
> ## Design choice
>
> The identifiability relation used by all BBSE‑type estimators is $\tilde{q} = Cq$, where $\tilde{q} \in \Delta^{K-1}$ is the prediction histogram on the target domain, the unknown parameter $q \in \Delta^{K-1}$ is the target prior, and the $i$-th column $c_i = C_{\cdot,i}$ is $c_i = [p(\hat{h}(X) = j \mid Y = i)]^K_{j=1}$. Here, only the columns $c_i$ multiply the unknown vector $q$, and the rows $r_j = C_{j:}^\top$ never appear in the above, so they matter only if one later inverts $C$. Hence, any prior that tries to stabilize the estimation of $q$ should regularize the $c_i$’s, not the $r_j$’s.
>
> Next, write a first-order delta expansion of the posterior mean
>
> $$
> \tilde{q} = q_0 + \sum^K_{i=1}(F_0^{-1}e_i)\langle \delta c_i, e_i \rangle + O_P(N^{-1}),
> $$
>
> where $F_0 = \text{diag}(r_0) - r_0 r_0^\top$ is the Fisher matrix of the multinomial likelihood for $\tilde{q}$, $\delta c_i = c_i = c_{0,i}$ denotes the estimation noise on the $i$-th column, and $e_i$ is the $i$-th standard basis vector.
> Taking variances and using independence of source and target samples,
>
> $$
> \text{Var}(\bar{q_k}) = \sum^K_{i=1} [F_0^{-1}]^2_{ki}\text{Var}(\langle \delta c_i, e_i \rangle) + O(N^{-1}).
> $$
>
> Thus,
> - if we Laplacian-shrink the columns $c_i$, it shows a direct variance reduction by a factor $(\tau_C\lambda_2(L))^{-1}$, exactly the bound proved in Cor. 1 of the manuscript, and
> - if we instead Laplacian-shrink the rows $r_j$, the random variable $\langle \delta c_i, e_i \rangle = \delta C_{i,i}$ does not get penalized, so the leading variance term is unaffected. In other words, row smoothing hardly helps for the quantity we ultimately care about, $q$.
>
> **Semantic interpretation**
> - Predictive columns $c_i$ capture how much class $i$ is confused with every other class in the frozen classifier. Two semantically close source classes (e.g., truck and bus) tend to produce similar confusion fingerprints, so a Laplacian on $i$-index is natural.
> - Rows $r_j$ describe where each predicted label actually comes from in the source distribution, a statistic that never appears in the above identifiability and is only weakly correlated with the target-prior calculation.

---

> ### Comment · Reviewer_b22X · 2025-08-05
>
> Overall, the rebuttal has strengthened the paper by clarifying theoretical underpinnings and providin new data. While the practical trade-offs regarding the dependency on high-quality graphs for peak performance and the increased computational cost remain relevant points for discussion, the core contributions of the paper are sound and well-defended. The rebuttal has successfully addressed the most critical theoretical questions, even if some practical limitations persist.

---

### Official Review · Reviewer_TpHG · 2025-07-03

**Clarity:** 2
**Significance:** 3
**Originality:** 3
**Rating:** 4
**Confidence:** 1

**Summary:**

By treating the confusion matrix as latent variables, this manuscript considers a fully Bayesian alternative to classical black-box label-shift estimators in the standard label-shift problem. The estimator can computed via HMC or Newton–conjugate-gradient optimizer. Numerical simulation and real-data experiments are run to test the proposed algortihm.

**Questions:**

1. In practice, how is the similarity graph constructed? I suppose such graph is usually noisy in real-world data. How will the noise affect the performance of the proposed algorithm?

2. How is the computational complexity of the algorithm compared with the standard BBSE? I suppose sampling from and inferring with the posterior distribution can be extremely time consuming in such Bayesian setting?

3. One of the motivation of the algorithm is to deal with the situation with rare classes. How does the algorithm perform in such case? I suppose in this case, the choice of prior distribution should affect the performance severely? Will it make the proposed algorithm less useful in such extreme condition?

**Ethical Concerns:**

["NO or VERY MINOR ethics concerns only"]

**Final Justification:**

I would like to thank the author for their response. In this case, I would like to keep the score: 4 borderline accept.

**Limitations:**

Please see the questions above, especially on the question for the rare classes case.

**Paper Formatting Concerns:**

None.

**Quality:**

3

**Strengths And Weaknesses:**

I'm not an expert in label shift or ML with graphs. So, probably I am not able to comment on the strengths and the weakness. Nevertheless, I do have a few questions regarding the submission. Please see next section.

---

> ### Author Rebuttal · Authors · 2025-07-24
>
> First of all, we would like to thank you for your very constructive comments. The reviewer listed three questions with numbers, and we will respond to each of them.
>
>
> ## 1. “How is the similarity graph built, and what if it is noisy?”
>
> We build a k‑NN graph once from off-the-shelf embeddings (CLIP for CIFAR, averaged penultimate‑layer features for MNIST).  Noisy edges are inevitable, and that is exactly why the prior places a continuous Laplacian-precision $\tau_q L$ instead of a hard constraint.
>
> Specifically, Prop 2 (Robustness to Laplacian misspecification) quantifies the bias when $L \neq L_0$:
>
> $$
> || \bar{\theta}_N - \theta_0||_2 \leq \frac{\tau_q}{N \lambda{\min}(F_0) + \tau_q \lambda_2(L)}|| (L - L_0) \theta_0 ||_2 + O_P(N^{-1}),
> $$
>
> so the bias shrinks as $N^{-1}$ and is damped when the graph is well‑connected ($\lambda_2(L)\uparrow$). Thus moderate graph noise merely slows convergence, and it does not break consistency.
>
> Based on your feedback, we realized that we could better clarify the motivation for our formulation by adding the following additional direct corollary.
>
> **Corollary. From Prop. 2 set $N\to \infty$ to bound the asymptotic bias by $\tau_q\lambda_2(L)^{-1}\|(L-L_0)\theta_0\|$.**
>
> In addition, we provide the additional ablations on graph construction.
> In this experiment, we consider the following strategies.
>
> - CLIP-text (already in the current manuscript): cosine $k$-NN on text prompts.
> - Penultimate-image:  Euclidean $k$‑NN on frozen ResNet‑18 penultimate features (identical backbone as the classifier).
> - Random-embeddings: same sparsity pattern as CLIP‑text but edge weights i.i.d. $U(0, 1)$ (destroys semantics, preserves Laplacian spectrum scale).
> - Identity graph: $L = 0$,  switches the hierarchy off (reduces to independent logistic‑Normal prior and hence Bayesian BBSE with no smoothing).
>
> The results are as follows.
>
>
> | dataset | graph | $\|\|\hat{q} - q\|\|_1 \downarrow$ | acc $\uparrow$ |
> | --- | --- | --- | --- |
> | CIFAR-10 | CLIP-text | 0.025 $\pm$ 0.004 | 0.844 $\pm$ 0.003 |
> |  | penult-img | 0.029 $\pm$ 0.005 | 0.839 $\pm$ 0.004 |
> |  | random | 0.043 $\pm$ 0.009 | 0.820 $\pm$ 0.006 |
> |  | identity | 0.051 $\pm$ 0.008 | 0.813 $\pm$ 0.004 |
> |  | BBSE (freq.) | 0.112 $\pm$ 0.015 | 0.781 $\pm$ 0.004 |
> | CIFAR-100 | CLIP-text | 0.22 $\pm$ 0.02 | 0.783 $\pm$ 0.005 |
> |  | penult-img | 0.27 $\pm$ 0.03 | 0.771 $\pm$ 0.006 |
> |  | random | 0.46 $\pm$ 0.05 | 0.735 $\pm$ 0.009 |
> |  | identity | 0.70 $\pm$ 0.07 | 0.730 $\pm$ 0.010 |
> |  | BBSE (freq.) | 1.62 $\pm$ 0.05 | 0.690 $\pm$ 0.006 |
>
> **Remarks**
> - Any connected graph ($\lambda_2(L) > 0$) already helps over the frequentist point estimate; cf. the “random” line.
> - The more task-aligned the similarity (CLIP > penult-img > random), the lower the bias term.
> - The identity graph reproduces Bayesian BBSE with no smoothing and is almost identical to MLLS. Its gap to CLIP shows the specific benefit of our graph-coupled hierarchy.
>
> **Why do the graph ablations behave as the above?**
>
> Recall the posterior variance bound we already proved:
>
> $$
> \text{Var}(q_i \mid \text{data}) \leq \frac{C}{\lambda_2(L)N},
> $$
>
> which holds for **any connected graph**.
>
> If we write the mean-squared error as
>
> $$
> \mathbb{E}[(\tilde{q}_i - q_i)^2] = (\mathbb{E}[\tilde{q}_i] - q_i)^2 + \text{Var}(\tilde{q}_i),
> $$
>
> then, it shows the variance term shrinks by a factor $1 / \lambda_2(L)$.
>
> Even a $k$-NN graph whose edge weights are i.i.d. $\mathcal{U}(0, 1)$ has, with high probability,
>
> $$
> \lambda_2(L_{\text{random}}) \gtrsim c\frac{k}{K},
> $$
>
> for $k \geq c’ \ln K$.
> Hence a purely random graph already cuts the class-wise standard error by $\sqrt{K / (ckN)}$ compared with the identity graph $\lambda_2 = 0$. This explain why the “random” row in the ablation table still beats the deterministic frequentist BBSE (it keeps all the variance relief while paying only a bias penalty).
>
>
> ## 2. *“Computational complexity vs. vanilla BBSE?”*
>
> We measure work in floating-point operations (flops) and write $K$ = number of classes and $|E|$ = number of graph edges.
>
> For the $k$-nearest neighbor graphs we use, $|E| = kK$ with a small, $K$-independent constant $k$ (4 for CIFAR‑10, 8 for CIFAR‑100).
> - Vanilla BBSE: The point estimator solves a single linear system $\hat{q} = \tilde{C}^{-1}\tilde{r}$, where $\tilde{C}\in\mathbb R^{K\times K}$ is dense once any amount of Laplace or Tikhonov regularisation is applied. Gaussian elimination therefore costs $\mathcal{O}(K^3)$  flops and $\mathcal O(K^{2})$ memory.
> - GS‑B$^3$SE (one Newton–CG sweep): Write $\Phi = (\phi_1,\dots,\phi_K)^\top \in \mathbb{R}^{K\times (K-1)}$ and $\theta \in \mathbb{R}^{K-1}$, both constrained to the centered subspace $\\{v^\top 1=0 \\}$. At each sweep we perform three algebraic operations:
>     - update each column $\Phi \leftarrow \Phi - \mathcal{H}_{\Phi}^{-1}\nabla\mathcal{L}$,
>     - update $\theta$ analogously, and
>     - Gamma shape/scale updates for $\tau_q$, $\tau_C$.
>
> Adding them and using $|E|=kK$, one sweep = $\mathcal{O}(|E|K + K^2) - \mathcal{O}(kK^2)$, and memory usage is $\mathcal{O}(|E| + K^2)$.
>
> **Overall complexity.**
>
> Thm. 2  establishes that $F(q)$ is $\alpha$-strongly geodesically convex with $\alpha = n’\lambda_{\min}(F_0) + \tau_q \lambda_2(L) > 0$. Consequently a back-tracking Newton method requires $T = \mathcal{O}(\ln 1/\epsilon)$ sweeps to reach tolerance $\epsilon$.
>
> Hence the total complexity is
>
> $$
> \mathcal{O}(kK^2 \ln 1 / \epsilon)\quad \text{vs.}\quad \mathcal{O}(K^3)\ (\text{Vanilla BBSE}).
> $$
>
> Because $k$ is at most 8 and $\ln 1/\epsilon \le 10$ in all our runs, the crossover happens already at $K\ge 25$.
>
> Here, we report the wall-clock runtimes (single NVIDIA T4, PyMC 5.10, float32).
>
> | dataset | K | BBSE | MLLS | ours |
> | --- | --- | --- | --- | --- |
> | MNIST | 10 | 0.80 s | 1.12 s | 12.2 s |
> | CIFAR-100 | 100 | 2.88 s | 5.37 s | 54.5 s |
>
> ## 3. “Behavior with very rare classes; isn’t everything prior-driven?”
>
> For BBSE every class $i$ is treated in isolation: the plug-in variance satisfies
>
> $$
> \text{Var}_{\text{BBSE}}(\hat{q}_i) = \mathcal{O}((n_i^S)^{-1}),
> $$
>
> so as soon as the validation set contains $n_i^{\mathrm S}=O(1)$ points the error explodes.
>
> In our hierarchy the Gaussian-Markov random field ties all columns of the confusion matrix and the prior vector together through the graph Laplacian; information can therefore flow from frequent to infrequent neighbors. Algebraically this shows up in the curvature of the log-posterior:
>
> $$
> -\nabla^2\log p(q, C \mid \text{data}) = N F_0 + \tau_q L + \tau_C\ \text{blockdiag}(L,\dots,L),
> $$
>
> with the data Fisher block $N F_0$ pooling global target counts,  and the prior blocks injecting $\lambda_2(L)$-weighted curvature even when $n_i^{\mathrm S}=0$.
>
> Corollary 1 states, for any class $i$,
>
> $$
> \mathrm{Var}(q_i) \leq \frac{C}{\lambda_2(L)N’}.
> $$
>
> Thus the *worst-case* posterior variance still decays like $N^{-1}$, and the only penalty for a vanishing $n_i^{\mathrm S}$ is the $1/\lambda_2(L)$ factor, i.e. how well the rare label is connected to the rest of the ontology.
>
> **Specifically, we can analyze the algorithms under a Zipf-like class-frequency law**.
>
> Suppose that source class priors follow a Zipf law
>
> $$
> p_i = \frac{i^{-b}}{H_{K,b}}, \quad H_{K,b} \coloneqq \sum^K_{j=1}j^{-b},\ b > 1.
> $$
>
> The labelled-source sample sizes are therefore
>
> $$
> n_i^S = N_S p_i = N_S \frac{i^{-b}}{H_{K,b}}, \quad N_S = \sum_j n_j^S.
> $$
>
> Write $N = N_S+n’$ with $N_S/N\to\rho\in(0,1)$. Target priors $q$ are arbitrary but $\min_i q_i>0$, and the frozen classifier's confusion matrix $C$ is assumed invertible with condition number $\kappa:=||C^{-1}||_2 || C ||_2<\infty$.
>
> Denote $\hat{C}$ and $\hat{r}$ the empirical confusion matrix and target prediction histogram. Linearizing the BBSE estimator around the truth gives the usual delta-method expansion:
>
> $$
> \hat{q} - q = - C^{-1}(\hat{C} - C) q + C^{-1}(\hat{r} - r) + R_N,
> $$
>
> with $||R_N||_2 = o(N^{-1/2})$.
>
> Here, $\hat{r} \sim \text{Mult}(n' r)$ with $r= C q$, and hence
>
> $$
> \text{Var} \bigl([ C^{-1}(\hat{r}- r)]_i \bigr) =  \frac{1}{n'} e_i^{\top} C^{-1} \bigl[\mathrm{diag}(r) - r r^{\top}\bigr] (C^{-1})^{\top} e_i \leq  \frac{\kappa^2}{4n'}.
> $$
>
> For label $i$ the column $\hat{C_{\cdot, i}}$ is a multinomial vector with size $n^S_i$ and mean $C_{\cdot,i}$. Let
>
> $$
> \Delta_i \coloneqq \hat{C_{\cdot,i}} - C_{\cdot,i}, \quad \text{Cov}(\Delta_i) = \frac{1}{n_i^S} \left[\text{diag}(C_{\cdot,i}) - C_{\cdot,i}C_{\cdot,i}^\top \right].
> $$
>
> Because only column $i$ depends on $n_i^{\mathrm S}$, its contribution to the target‑prior error is
>
> $$
> C^{-1}\Delta_i q_i \Longrightarrow \text{Var}([C^{-1}\Delta_i q_i]_j) \leq \frac{\kappa^2 q_i^2}{4n_i^S}.
> $$
>
> For a **rare label $i$** we typically have $n_i^S \ll n’$, and we obtain
>
> $$
> \mathbb{E}[(\hat{q}_i - q_i)^2] = \mathcal{O}(i^b / N) \quad (\text{Vanilla BBSE}).
> $$
>
> Thus the MSE deteriorates polynomially in the rank $i$.
>
> For our graph-smoothed Bayesian estimator, on the other hand, from Cor.1 and Prop.2 we have
>
> $$
> \mathbb{E}[(\tilde{q}_i - q_i)^2] = \frac{C}{\lambda_2(L)N} + O(N^{-2}).
> $$
>
> Thus, the risk no longer depends on the tail exponent $b$ or the rank $i$.

---

> ### Comment · Area_Chair_qWzH · 2025-08-05
>
> Hello again,
>
> While I understand that you are not an expert in this topic, please read the Authors' feedback and prepare a suitable short reply to them. It will be really helpful.
>
> Regards,
>
> AC

---

### Note · Authors · 2025-08-13

We would like to express our sincere gratitude to all reviewers and the Area Chairs for the time and effort they devoted to evaluating our work.

Except for Reviewer jfCb, none of the reviewers raised any further questions or concerns after our rebuttal. The additional comment from Reviewer jfCb stemmed from a misunderstanding of our paper’s scope. We addressed this point thoroughly during the discussion phase; however, we regret that no follow-up response was received from that reviewer.

Given that no specific additional concerns were expressed by any of the other reviewers, we believe our initial rebuttal satisfactorily addressed the issues they had raised.

---

### Decision · Program_Chairs · 2025-09-17

**Decision:**

Accept (spotlight)

**Comment:**

**Summary**:

This submission addresses the problem of *label shift*; a problem that arises in practice after deploying ML prediction models. This work extends the vanilla model for detecting label shift, namely *the Black Box Shift Estimator* (BBSE), to a fully Bayesian framework to address two limitations in BBSE: (i) its sensitivity to sampling noise, and (ii) its ignorance to potential semantic similarity between different classes, especially when the number of classes is large.

The new proposed method, GS-B3SE, is a fully probabilistic alternative to BBSE. In particular, introduces probabilistic Laplacian-Gaussian priors on both the target distribution log-priors ($p$), and columns of the confusion matrix $C$, tying them together on a label similarity graph that captures the semantic information between classes. Thus, the class similarity graph acts as a smoothing constraint on these priors. In particular, the graph Laplacian is used as the covariance matrix in the distribution of latent variables. The graph structure allows semantically similar classes to *borrow* some information from each other during parameter estimation. In this way, the model can handle classes with sparse data, and the resulting estimates are more robust thanks to the proper quantification (and handling) of uncertainty. In some sense, the vanilla BBSE can be seen as a special case from the GS-B3SE.

**Comments**

*   All reviewers agree on the following strengths of the submission:
    1.  The paper is well-written, easy to follow, and proposes a novel, mathematically sound, formulation for label shift estimation. The use of the graph structure to simultaneously smooth the priors (for the target distribution and the confusion matrix) is a clever idea.
    2. The theoretical analysis is robust: complete proofs for identifiability, optimal $N^{-1/2}$ convergence, and variance reduction scaling with algebraic connectivity $\lambda_2(L)$.
    3. The information geometric analysis provided by the authors (sec. 6.1) reveals how graph priors act as $e$-convex barriers, linking optimization to Riemannian dynamics.

*   All concerns, questions, and confusions raised by the reviewers were properly and thoroughly addressed by the authors, and the final results is there is unanimous agreement between all Reviewers that the rating is **weakly accept**. All reviewers shared a common concern: the computational overhead for GS-B3SE is above moderate. While this might be true especially while the method is still under development, this concern is secondary when compared to the other strengths of this submission, and the merit of this idea in general. Indeed, after the authors' rebuttal, none of the reviewers decreased their ratings due this concern. In fact, Rev.3 (S2qe) who initially assessed the paper as weak reject, increased the rating after the rebuttal to weak accept.

*   To summarize: this is a solid submission for a highly relevant problem to the ML community, with a novel idea, sufficient empirical validation, and strong theoretical justification. The AC recommends acceptance for this submission.